

# Uncertainty quantification for overshoots of tipping thresholds

Kerstin Lux-Gottschalk[1,*] and Paul D. L. Ritchie[2,3,*]

[1]Eindhoven University of Technology

[2]Department of Mathematics and Statistics, Faculty of Environment, Science and Economy, University of Exeter, North Park Road, Exeter, EX4 4QE, UK

[3]Global Systems Institute, Faculty of Environment, Science and Economy, University of Exeter, North Park Road, Exeter, EX4 4QE, UK

[*]These authors contributed equally to this work.

**Correspondence:** Kerstin Lux-Gottschalk (k.m.lux@tue.nl) and Paul D. L. Ritchie (Paul.Ritchie@exeter.ac.uk)

**Abstract.** Many subsystems of the Earth are at risk of undergoing abrupt transitions from their current stable state to a drastically different, and often less desired, state due to anthropogenic climate change. These so-called tipping events often present severe consequences for ecosystems and human livelihood that are difficult to reverse. One common mechanism for tipping to occur is via forcing a nonlinear system beyond a critical threshold that signifies self-amplifying feedbacks inducing tipping.

However, previous work has shown that it is possible to briefly overshoot a critical threshold and avoid tipping. For some cases, the peak overshoot distance and the time a system can spend beyond a threshold are governed by an inverse square law relationship Ritchie et al. (2019). In the real world or complex models, critical thresholds and other system features are highly uncertain. In this work, we look at how such uncertainties affect the probability of tipping from the perspective of uncertainty quantification. We show the importance of constraining uncertainty in the location of the critical threshold and the linear restor-

ing rate to the system's stable state to better constrain the location of the boundary separating overshoots that avoid tipping from those that do not. We first prototypically analyse effects of an uncertain critical threshold location separately from effects due to an uncertain linear restoring rate. We then perform an analysis of joint effects of uncertain system characteristics within a conceptual model for the Atlantic Meridional Overturning Circulation (AMOC). The simple box model for the AMOC shows that these uncertainties have the potential to reverse conclusions for mitigation pathways. If the uncertain critical threshold

were to be further away than previously considered, a pathway that may have been in danger of tipping, may no longer involve an overshoot at all. In our study, we highlight the need to constrain the highly uncertain diffusive timescale (representative of wind-driven fluxes) within the box model to reduce tipping uncertainty for overshoot scenarios of the AMOC.

## 1 Introduction

Climate Action is listed as one of the United Nations´ 17 sustainable development goals[1]. This includes climate tipping points,

which have gained increasing attention by scientists, public and policymakers Lenton et al. (2023). Tipping events are sudden transitions that may occur when environmental conditions exceed critical thresholds Scheffer et al. (2012). Systems may be kicked off their current stable state to a drastically different state, which can be irreversible Lenton et al. (2008). These tipping

---

[1]source: https://sdgs.un.org/goals, last accessed: December 19, 2023.





points pose severe threats to ecosystems and human habitat Lenton et al. (2023). The evolution of many subsystems of the Earth, such as the Atlantic Meridional Overturning Circulation (AMOC), are prone to exhibit tipping behaviour instead of only featuring small gradual changes Armstrong McKay et al. (2022).

Awareness of the need for action is prevalent since the impact is likely to be far reaching if a system tips, see e.g. Ritchie et al. (2020). To increase the efficiency of measures, a better understanding of these tipping events and under which circumstances tipping can be prevented is crucial. For some subsystems of the Earth, critical thresholds are assumed to be at low levels of global warming Armstrong McKay et al. (2022), such that overshoots of the threshold are becoming increasingly likely. It is important to note that, for some elements, tipping can still be avoided under a sufficiently fast reversal of the forcing Jackson et al. (2022). This overshoot mechanism has already been subject to thorough investigations Ritchie et al. (2021); O'Keeffe and Wieczorek (2020); Wunderling et al. (2023); Bochow et al. (2023). However, much less is known about the impact of uncertainties on overshooting tipping thresholds, i.e. uncertainties in a climate model regarding choices of model parameter values Lux et al. (2022). These uncertainties might significantly affect the mitigation window, that is how far and for how long a system may overshoot tipping thresholds and still retain the system's original equilibrium state Ritchie et al. (2019). There is a need for further research on overshoots of tipping thresholds to quantify the mitigation window more narrowly since tipping events are associated with large uncertainties about the degrees of global warming that would cause the system to tip. Since for some systems we are already very close to a level of global warming that might trigger tipping (see (Armstrong McKay et al., 2022, Figure 2)), it is crucial to quantify which exceedance level of a possible threshold might allow us to return to the original state if forcing is reversed sufficiently quickly.

Therefore, in this work, we focus on the quantification of uncertainty in overshooting tipping thresholds resulting from uncertainty in system characteristics for a given forcing profile. In particular, we illustrate our methodology for one of the most prominent tipping elements, the AMOC, where the related uncertainty about the threshold level of warming is particularly pronounced Armstrong McKay et al. (2022).

A tipping of the AMOC would be likely to cause a significant cooling over Northern Europe, substantially change patterns in tropical rainfall, as well as trigger regional sea level rise Jackson et al. (2015). Gaining a better understanding of the tipping behaviour of the AMOC is crucial for deriving efficient climate change mitigation strategies. There already exist hosing experiments studying whether the AMOC recovers or not after overshooting a critical threshold Jackson and Wood (2018a); Jackson et al. (2022). In addition to those, here, we investigate AMOC overshoot scenarios with the aim of gaining a conceptual understanding of the mechanisms involved and in particular how uncertainty affects the mitigation window. This is not only important for the AMOC, but for many other systems as well Ritchie et al. (2021); Meyer et al. (2022); Bochow et al. (2023). Further so called tipping elements of the climate system have been identified in Lenton et al. (2008). A recent assessment can be found in Armstrong McKay et al. (2022), where the authors elaborate on the most important tipping elements and corresponding tipping points.

To understand overshoots of tipping thresholds, it is important to understand which mechanisms can cause a system to tip. We distinguish between Ashwin et al. (2012)

1. bifurcation-induced tipping,



2. noise-induced tipping, and

3. rate-induced tipping.

For mechanism 1., an environmental parameter that exceeds a critical threshold, the bifurcation value, causes the system to tip. In Kuehn (2013), the authors provide a mathematical framework for critical transitions in terms of bifurcation theory. More recently, the universal nature of the emergence of critical transitions in physical systems has been analyzed in Kuehn and Bick (2021). For further details on the theory of bifurcations, we refer the reader to Kuznetsov (2004); Wiggins (2003). Environmental fluctuations that become particularly pronounced might also cause a system to tip, known as noise-induced
tipping (mechanism 2.), see e.g. Ashwin et al. (2012); Ma et al. (2019). In particular, such a transition can already happen before the environmental forcing parameter exceeds its corresponding threshold or even in a setting where there is no inherent bifurcation at all. Last but not least, also the rate at which environmental/system parameters change matters. In mechanism 3., a fast enough change can also trigger a so-called rate-induced tipping event. For a recent overview on rate-induced tipping, we refer the reader to Ritchie et al. (2023). Moreover, the different mechanisms of tipping may interact with each other.
For example, studies Ritchie and Sieber (2017); Slyman and Jones (2023) have analysed the interplay between rate- and noise-induced tipping. The combination of rate- and bifurcation-induced tipping has already been addressed as well, see e.g. O'Keeffe and Wieczorek (2020); Alkhayuon et al. (2019).

Although all three mechanisms can contribute to the uncertainty in mitigation windows, here, our primary focus is on bifurcation-induced tipping. We consider various profiles for the change in the environmental parameter that have different
impacts on the overshoot of the critical bifurcation (threshold) value. Most importantly, we analyse these overshoots in the presence of uncertainties in model parameters. Taking these uncertainties into account is crucial for a better understanding of safe mitigation windows in real world climate systems.

In particular, the IPCC 2021 report Masson-Delmotte et al. (2021) emphasises on high impact, low likelihood climate outcomes, to which some tipping events belong, as being part of climate risk assessments. To assess risks of tipping, a thorough
handling of uncertainties is needed. One type of uncertainty related to a possible tipping of the AMOC is uncertainty in datasets such as in sea-surface temperature and salinity datasets. An uncertainty propagation procedure to quantify effects of dataset uncertainties on indicators of critical slowing down has recently been proposed in Ben-Yami et al. (2023). Here, we focus on uncertainties in model parameters of a conceptual AMOC model. These uncertainties affect the time and the peak overshoot distance that would still facilitate a return to the original state. More precisely, we use the Stommel-Cessi model Cessi (1994) to
conceptually illustrate the far reaching effect of uncertainty in wind-driven gyres and eddies, represented by the diffusive time scale parameter, on mitigation windows that avoid tipping. This model exhibits a double-fold bifurcation and thus includes a range of forcing parameters where the AMOC exhibits multistability. The AMOC model equation including the diffusive time scale parameter of interest is given by (A10) and a conceptual equation to illustrate inherent characteristics of a fold bifurcation is provided in (A7) in the Methods Section.
The aim is to develop a probabilistic extension of the work of Ritchie et al. (2019). Therein, the authors derived an inverse-square law between characteristics determining a mitigation window in case of a fold tipping scenario. The condition provided



in (1) specifies the mitigation window in terms of the exceedance time over the threshold $t_{over}$ and the peak external forcing $p_{peak}$ over the critical threshold $p_b$

$$(p_{peak} - p_b)t_{over}^2 < \frac{4}{a_0^2 \kappa}, \tag{1}$$

where $\kappa$ is proportional to the linear restoring rate, which is a measure of the recovery rate back to the equilibrium after a perturbation is made, and $a_0$ is the inverse of the system's timescale. Hence, to be within the mitigation window, the product of the squared time spent over the threshold and the peak overshoot distance needs to be smaller than a quantity that depends on system specific parameters, which can be highly uncertain in real-world applications. For example, the critical threshold, $p_b$, and the linear restoring rate, proportional to $\kappa$, might not be known exactly.

The paper is structured as follows: First, in the *Results* section we present how uncertainty in system parameters affects the mitigation window specified in equation (1). The results section is subdivided into three subsections where the first two address how uncertainty affects the mitigation window in the prototypical fold bifurcation setting, inherent to many conceptual climate models. Thereby, we distinguish between uncertainty in (i) the location of the tipping threshold $p_b$ and (ii) the linear restoring force proportionality constant $\kappa$. The third subsection provides results on the uncertainty in the mitigation window for a conceptual AMOC model, the Stommel-Cessi box model Cessi (1994), which exhibits two of these fold bifurcations. The uncertain diffusive timescale in this model entails uncertainty in both $p_b$ and $\kappa$, thus exhibiting effects showcased in the previous two subsections. In the *Discussion* section, we expand on how results on the probability of tipping for the presented scenarios might inform decisions about alternative mitigation pathways. Further details of the forcing profiles, overshoot theory, model configurations and Bayesian inference method can be found in the *Materials and Methods* section.

## 2 Results


We begin by considering a simple conceptual model for tipping via the prototypical fold bifurcation. Utilising such a simple model allows us to isolate how uncertainty in either the location of the tipping threshold or the linear restoring force of the system affects the probability of tipping. We illustrate these effects graphically by showing how a fixed exemplary overshoot profile can result in vastly contrasting outcomes based on different realisations of uncertain model parameters (**tipping threshold location or linear restoring force**). Finally, for given probability distributions of the model parameters, we illustrate how the probability of tipping varies for different overshoot profiles characterised by the peak external forcing level and time over a prescribed threshold.

### 2.1 Uncertain location of tipping threshold

Arguably, the biggest uncertainty to consider is the location of the tipping threshold. For a given overshoot profile, the threshold
location will ultimately determine if an overshoot of the critical tipping threshold (later referred to as just threshold) occurs or not. Clearly, without variability in the system, no tipping will occur if no overshoot of the threshold takes place regardless of





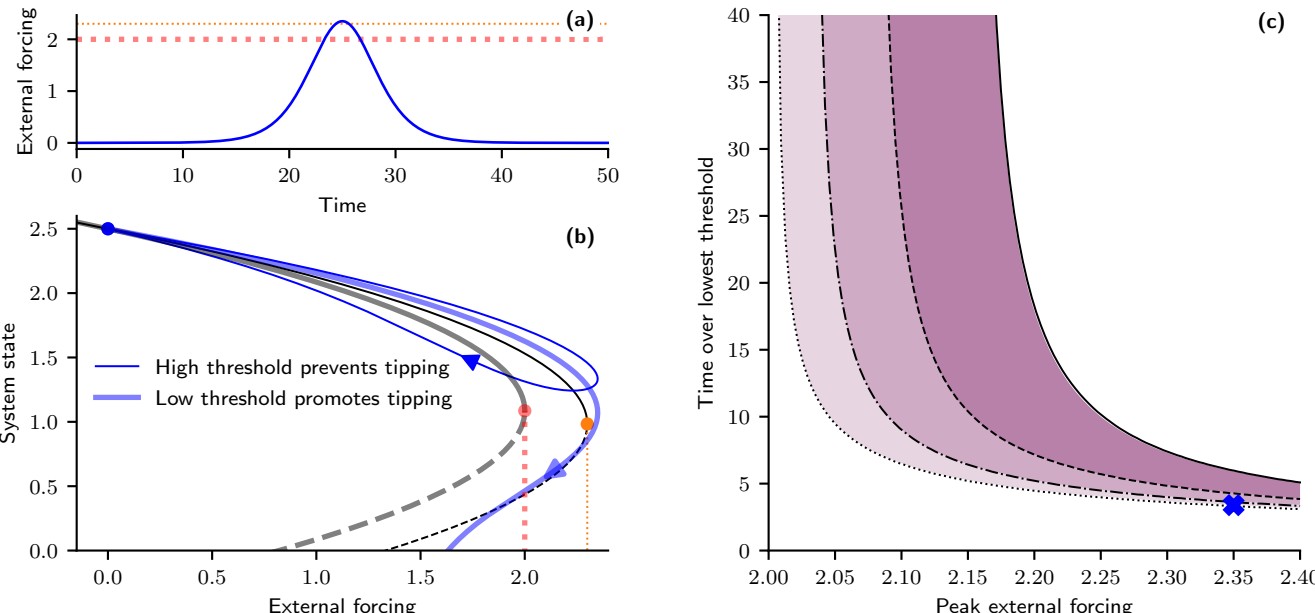

**Figure 1. Probabilistic overshoots given uncertainty in location of tipping threshold.** (a) Time profile of external forcing briefly overshooting an uncertain tipping threshold assumed to lie between the red (low threshold) and orange (high threshold) dotted lines. (b) System responses (blue) subjected to the external forcing profile given in (a). System with high threshold (thin and opaque) avoids tipping, whereas, the same system but with an early threshold (thick and translucent curves) undergoes tipping. Steady states indicated by black curves, are either stable (solid) or unstable (dashed). Orange and red dots indicate threshold location (fold bifurcation) of the respective systems. (c) Probabilistic critical boundaries separating overshoots that avoid tipping from those that do not, in the plane of time over the lowest threshold (red dotted line in (a) and (b)) and peak forcing amplitude, given a uniform distribution in the threshold location. Purple colour gradient shows different probability boundary levels derived from the theory. Specifically, from left to right: the start of the lightest purple region indicates the location of the 1% probability of tipping critical boundary; the medium shade 10%; and the darkest shade starts at 25%; and finishes at 50%. Black curves provide the exact boundaries calculated numerically (dotted – 1%, dash-dot – 10%, dashed – 25%, solid – 50%). Blue cross corresponds to time profile of external forcing given in (a).

how long it takes to reverse the forcing. Additionally, if an overshoot of the threshold does occur, the threshold location will determine how long that overshoot lasts for. We now illustrate and develop this idea further in Figure 1.

We consider a single forcing profile, which we assume to be given, that starts at some initial level of forcing, smoothly increases to a peak level before returning back to the initial level at a mirrored rate to the approach, as shown in Figure 1(a). Note again that the location of the tipping threshold will determine how large (if any) and how long the system will be beyond the tipping threshold for this single forcing profile. If the threshold is low (red dotted line) then the overshoot will be large and long whereas a higher threshold (orange dotted line) means that the overshoot will be smaller and for less time.

The contrasting consequences of a system having either a low or high threshold are demonstrated in Figure 1(b). The thick 130 translucent curves correspond to a system with an early threshold, which causes the system to tip due to the large and long




overshoot. However, for a high threshold, but otherwise identical system (thin and opaque curves), tipping is avoided for the same forcing profile since the overshoot is now comparatively small and for a short duration.

To summarize the effect of an uncertain location of the tipping threshold, we point out that a threshold further away would result in a smaller peak overshoot distance $p_{peak} - p_b$ and a shorter overshoot duration $t_{over}$. Therefore this minimises the two
terms on the left hand side of (1) making avoiding tipping more likely for the given overshoot profile.

We have seen that for a given overshoot forcing profile, tipping can either occur or be avoided depending on the location of the tipping threshold. If we have some initial estimate of the uncertainty for the location of the tipping threshold, then a tipping probability based on this distribution can be assigned for any forcing profile. Initially, we assume the location of the tipping threshold to be uniformly distributed between 2 and 2.3. The probability of tipping is plotted in Figure 1(c) for a range
of forcing profiles based on the time spent over the lowest threshold from this distribution and the peak in external forcing.

The purple shaded regions provide different critical probability boundaries as derived from the inverse square law theory (1) (with a small modification to account for time over an arbitrary threshold, see (A6) in Methods). The lower limit of the lightest shading gives the boundary for only 1% probability of tipping. The end of the lightest shading and start of the next shading corresponds to a 10% probability of tipping, the subsequent boundary 25% probability and the end of the darkest shading, 50%
probability. Meaning that below the shading, overshoots of the threshold are very unlikely ($< 1\%$) but above the shading with 50% probability and higher tipping will occur, for the assumed uncertainty in the location of the tipping threshold. We can also calculate the boundaries for the different probability levels from numerical simulations (see Methods for further details) and these are given by the black curves, ranging from dotted for the 1% level to solid for the 50% probability of tipping. These numerically calculated curves display a very good agreement to the theory especially for small and long overshoots.

For small peak levels in the external forcing there is a large uncertainty in the tipping behaviour. This is due to in some cases the profile not even overshooting the threshold, if the threshold is high, and therefore regardless of the time taken for the forcing to return, tipping will be avoided. Though for lower thresholds, there will be an overshoot and then the reversal in the forcing needs to be sufficiently quick otherwise tipping will occur. For larger peaks there will be an overshoot in most cases and therefore the uncertainty is reduced as there is a maximum time that the system can spend beyond the threshold before
tipping would ensue.

The blue cross corresponds to the overshoot profile given in panel (a), where the probability of tipping is close to the 1% boundary level for the range of tipping threshold locations considered. As previously discussed, the thick and translucent curves correspond to the lowest threshold and tipping is nearly avoided. Therefore, just having a slightly higher threshold would have prevented tipping, and for the highest threshold tipping is comfortably avoided (thin opaque curves).

If the uncertainty of the tipping threshold can be reduced then ultimately the uncertainty in the tipping behaviour can be constrained, as shown by Figure 2. Here we introduce a knowledge-based distribution, which is more constrained (for example through expert judgement or information inferred from data) than the initial, uninformed, uniform distribution. Figure 2(a) provides the comparison between the initial distribution in purple and the knowledge-based distribution, centred on a threshold location of 2.1, in green. The tipping probabilities for the two different threshold distributions are given in Figure 2(b).
Noticeably, the knowledge-based distribution corresponds to a much more constrained uncertainty in the tipping behaviour.





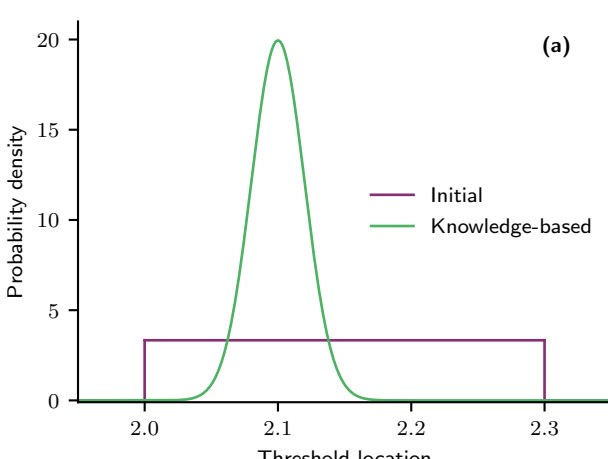
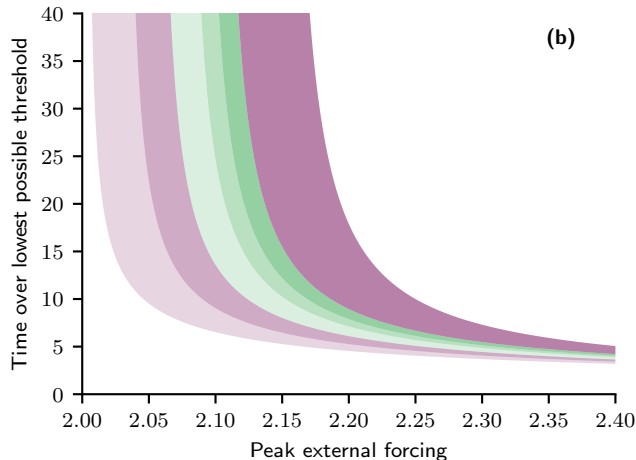

**Figure 2. Constraining uncertainty in threshold location minimises uncertainty in overshoot boundary separating tipping and avoiding tipping.** (a) Probability distribution functions for threshold location. A uniform distribution is used as the initial distribution (purple), whereas, the knowledge-based distribution is assumed to take the form of a normal distribution (green). (b) The theoretical probabilistic critical boundaries separating overshoots that avoid tipping from those that do not, in the plane of time over the lowest threshold and peak external forcing, are given in colour corresponding to the distributions given in (a). Specifically, from left to right: the start of the lightest shading indicates the location of the 1% probability of tipping critical boundary; the medium shade 10%; and the darkest shade starts at 25%; and finishes at 50%.

For example, forcing profiles that had a 10% probability of tipping for the initial distribution are now very unlikely ($< 1\%$) to occur given the knowledge-based distribution. Concurrently, the dark purple shaded region corresponds to where tipping was a likely occurrence ($25 - 50\%$) previously, but now tipping becomes more likely than not ($> 50\%$).

## 2.2 Uncertain linear restoring force

The previous section highlights how uncertainty in the location of the tipping threshold can affect the probability of tipping. However, another important factor for determining the mitigation window for overshoots is the strength of the linear restoring rate – decay rate towards the stable equilibrium. In this section, we use the same simple conceptual model, but keep the tipping threshold location fixed and vary the linear restoring rate instead.

Figure 3(a) provides an exemplary time series profile of an external forcing that starts below the tipping threshold (indicated

by the red dotted line), then increases such that there is a brief overshoot before reversing the forcing back to its original level. Similar to before, we can observe contrasting tipping behaviours for identical systems other than the strength of the linear restoring rate, see Figure 3(b). Namely, if the system has a weak linear restoring force the system will avoid tipping (thin and opaque blue curve). Whereas, if the restoring force is too strong tipping cannot be prevented. The example given by the thick and translucent curves nearly recovers but does not quite cross the unstable branch (representing the boundary for the basin of

attraction), and so ultimately tips due to the restoring force being too strong. The system with the weaker linear restoring force





has a weaker 'pull' towards the stable state and therefore lags further behind the equilibrium of the static system than that of the system with the stronger restoring force. Consequently, when the system is forced beyond the critical threshold the weaker system will in effect take longer to realise it's over the edge and runaway to an alternative state. Additionally, the curvature of the fold is less pronounced and so the unstable state will be crossed earlier (when reversing the forcing) for the weak system

(thin and opaque black dashed curve) than for the strong system (thick and translucent black dashed curve). All these factors culminate in the strong restoring system tipping and the weak restoring system avoiding tipping for the same forcing overshoot profile.

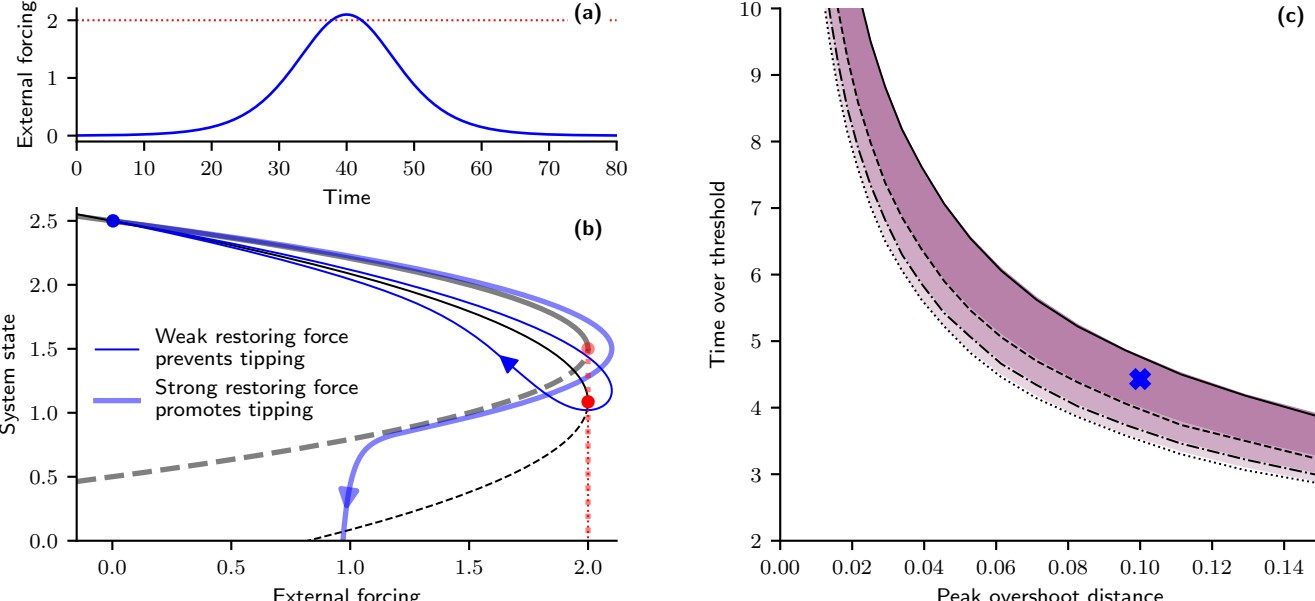

**Figure 3. Probabilistic overshoots given uncertainty in strength of linear restoring force.** (a) Time profile of external forcing briefly overshooting tipping threshold indicated by the red dotted line. (b) System responses (blue) subjected to the external forcing profile given in (a). System with weak restoring force (thin and opaque) avoids tipping, whereas, a system with stronger restoring forces (thick and translucent curves) undergoes tipping. Steady states indicated by black curves, are either stable (solid) or unstable (dashed). Red opaque and translucent dots indicate threshold location (fold bifurcation) of the respective systems. (c) Probabilistic critical boundaries separating overshoots that avoid tipping from those that do not, in the plane of the time over the threshold and peak overshoot distance, given a uniform distribution in the restoring force proportionality factor. Purple colour gradient shows different probability boundary levels derived from the theory. Specifically, from left to right: the start of the lightest purple region indicates the location of the 1% probability of tipping critical boundary; the medium shade 10%; and the darkest shade starts at 25%; and finishes at 50%. Black curves provide the exact boundaries calculated numerically (dotted – 1%, dash-dot – 10%, dashed – 25%, solid – 50%). Blue cross corresponds to time profile of external forcing given in (a).

Uncertainty in the linear restoring force of the system will again lead to uncertainty in the boundary separating overshoots that result in tipping from those that avoid tipping. Figure 3(c) provides a probabilistic assessment of the critical boundary





assuming an initial uniform distribution in a parameter determining the restoring force. As we assume the tipping threshold is known the axes now show peak overshoot distance and time spent over the threshold. As before the purple shading provides the probability intervals derived from the inverse square law theory and the black curves with different linestyles are the numerically calculated boundaries (1%, 10%, 25%, 50% from dotted to solid). The numerically calculated curves again display a very good agreement to the theory especially for small and long overshoots. The blue cross corresponds to the overshoot profile given in panel (a), where the probability of tipping is close to 50% based on the range of restoring force strengths considered.

Recall that for small peak external forcing levels the uncertainty in the tipping probability was large for uncertain tipping thresholds. In comparison for uncertain restoring forces the tipping uncertainty is greatly reduced (compare Figure 1(c) and Figure 3(c)). For a given trajectory if the threshold is uncertain, then there may be no overshoot of the threshold, meaning the time to reverse the forcing is irrelevant given tipping is not possible (without noise). Whereas, if only the restoring force is uncertain, then it is known if the trajectory overshoots the threshold. Therefore, if it does overshoot, only a limited time can be spent over the threshold before tipping ensues.

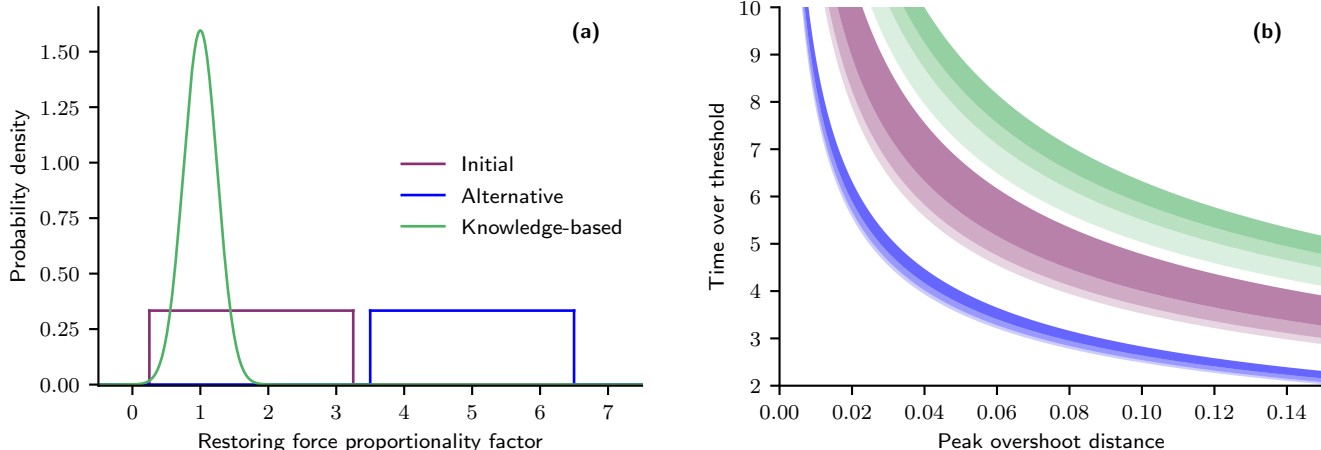

**Figure 4. Constraining uncertainty in linear restoring force does not necessarily reduce uncertainty in overshoot boundary separating tipping and avoiding tipping.** (a) Probability density functions for the restoring force proportionality factor. A uniform distribution is used as a prior (purple), whereas, the constrained posterior is assumed to take the form of a normal distribution (green). An alternative uniform prior distribution (blue) is also considered for same range of restoring force values but larger. (b) The theoretical probabilistic critical boundaries separating overshoots that avoid tipping from those that do not, in the plane of the time and peak distance over the threshold, are given in colour corresponding to the distributions given in (a). Specifically, from left to right: the start of the lightest shading indicates the location of the 1% probability of tipping critical boundary; the medium shade 10%; and the darkest shade starts at 25%; and finishes at 50%.

Figure 4 shows how changing the uncertainty in the restoring force affects the uncertainty in the probability of an overshoot avoiding tipping. In Figure 4(a) we again start with an initial uniform distribution (purple) and assume that the knowledge-based distribution (green) narrows down this uncertainty. The curves for the different probability levels of tipping, shift substantially,





see Figure 4(b). For a trajectory that sits on the initial 50% probability of tipping curve, would now be considered to be very unlikely ($< 1\%$) to cause tipping.

Interestingly, the separation between the 1% and 50% curves of the initial and knowledge-based distributions have barely changed. This seemingly counter-intuitive result can be explained by the change in the mean of the distribution counteracting the decrease in uncertainty. To illustrate this we consider an alternative uniform distribution (blue), that covers the same range but has a much higher mean for the restoring force, see Figure 4(a). The width of the banding covering the different probability levels, in Figure 4(b) is much smaller for this alternative uniform distribution than the initial uniform distribution. Indicating that an uncertainty in the restoring rate for large values is less critical than at lower values. Thus, when transitioning from the initial uniform distribution to the knowledge-based distribution, the reduction in uncertainty would narrow the banding but by decreasing the mean simultaneously also widens the banding. So, for this example, little change in the width is observed. Importantly though, the uncertainty of the location of the critical boundary (separating tipping from not tipping) does still decrease.

So far, we have illustrated the effects of uncertainty in the tipping threshold location and the linear restoring force separately. We now come to a joint analysis of uncertainty, by considering uncertainty in a model parameter for a simple conceptual model of the Atlantic Meridional Overturning Circulation (AMOC). The uncertainty stems from the diffusive timescale parameter that jointly influences the location of the tipping threshold and the linear restoring force.

## 2.3 Uncertain diffusive timescale in the Stommel-Cessi model

The model, introduced by Cessi (1994), is a modification of the 2-box Stommel model Stommel (1961), and describes the change in salinity flux – a proxy for the strength of the AMOC. If the freshwater flux added to the North Atlantic becomes too large, it is possible to exceed a critical threshold, represented by a fold bifurcation in the model. This would cause the AMOC to tip from its current "on state" to a collapsed state, if exceeded for too long. Both the location of this fold and the restoring rate are determined by the advective and diffusive (mixing by wind driven gyres and eddies) timescales. More details of the model can be found in the Methods section.

Figure 5 shows how the location of the critical freshwater fluxes (i.e. both the threshold indicating the transition to the collapsed off state and the threshold representing recovery back to the on state) and the width of the corresponding region of bistability changes with varying the advective and diffusive timescales. The regions plotted correspond to the plausible ranges for the two different timescales, as determined by Wood et al. (2019). The advective timescale is relatively well constrained, whereas the uncertainty in the diffusive timescale is much larger. Consequently, we fix the advective timescale to 70 years, and instead focus on the uncertainty in the diffusive timescale (indicated by the red dashed line).

Figure 5(a) shows the critical level of freshwater flux (denoted by colour), at which the AMOC on state terminates at a fold, based on the diffusive and advective timescales. The black lines provide contours of constant ratio between the diffusive and advective timescales. A sufficiently large ratio between the diffusive and advective timescales is required for the AMOC





to possess a region of bistability. Hence, below this critical ratio ($\eta^2 = 3$), no tipping is possible as there exists no critical freshwater flux and so the corresponding region is coloured white.

Previously, the critical non-dimensional freshwater flux was shown to only depend on the ratio of timescales and that increasing this ratio (either by decreasing the diffusive or increasing the advective timescales) moves the threshold earlier Lux et al. (2022). However, the scaling from dimensional to a non-dimensional freshwater flux, introduced by Cessi (1994), depends on the diffusive timescale, which we assume to be uncertain. Hence, the tipping threshold for the dimensional freshwater flux no longer remains constant along the contours of constant ratio. Instead, the tipping threshold in the dimensional freshwater flux

moves earlier for increasing either the advective or diffusive timescale, due to the scaling dependency of the diffusive timescale between the dimensional and non-dimensional freshwater flux. This can for example be seen by fixing the advective timescale at the red dashed line and observing that the critical freshwater flux decreases along this line as we increase the diffusive timescale.

     Similarly, this scaling dependency affects the location of the other fold bifurcation/threshold corresponding to the termi-
nation of the off state, see Figure 5(b). Lux et al. (2022) found that the alternative freshwater flux threshold (transition from AMOC off to AMOC on) was largely constant and independent of the ratio of timescales. Hence, for dimensional quantities this translates to the alternative threshold being independent of the advective timescale (i.e. for fixed diffusive constant the alternative threshold barely changes). On the other hand, the threshold decreases for increasing diffusive timescales – making it harder to restore the AMOC if it were to collapse.

The final panel, Figure 5(c), combines the first two panels by plotting in colour the difference between the two thresholds corresponding to the width of the region of bistability. Moving along the red dashed contour, decreasing the diffusive timescale decreases the width of bistability. Concurrently, the upper (and lower) critical threshold moves later, so these factors alone will make tipping less likely for any given overshoot. Additionally, the freshwater flux can be stabilised at a higher level such that only the AMOC on state exists.

We continue with analysing the AMOC tipping behaviour in the style of Figures 1 and 3. Here, in Figure 6(a) we consider a single freshwater flux profile that overshoots before stabilising at just below 0.25 $Sv$ ($1Sv = 10^6 m^3/s$). The system response to this overshoot trajectory is given in Figure 6(b) for both a small (thin and opaque curves) and large (thick and translucent curves) diffusive timescale. In the event that the timescale is small then the tipping threshold would be late (orange dotted lines), and so the overshoot would be small and for a short duration. Note also that the bistability region is small for small

diffusive timescales and therefore, in this example, the AMOC would be guaranteed to recover if the freshwater flux is reduced back to 0.2 $Sv$.

     In contrast, if the diffusive timescale is large, then the AMOC would collapse and not recover. This is by virtue of the threshold being a lot earlier (red dotted lines) causing the overshoot of the tipping threshold to be much larger and for a longer period of time. These combined factors, coupled with a larger bistability region (stabilising within the bistability region) mean

that the AMOC tips to its off state.

     The probability of avoiding tipping for different overshoot profiles characterised by the amount of time the freshwater flux is above the earliest threshold (based on the prior distribution for the diffusive timescale) and the peak freshwater flux is





visualised in Figure 6(c). For sufficiently small diffusive timescales, recovery of the AMOC on state is guaranteed even if the AMOC temporarily collapses. This is a result of the freshwater flux stabilising below the bistability region, where the on

state is the only stable equilibrium. For our prior uniform distribution, there is just under a 40% probability that the freshwater flux stabilises below the bistability region (but never above, which would guarantee tipping). Hence, we cannot rule out the possibility of AMOC recovery even for very large and long overshoots.

A good correlation is once again found between the inverse square law theory and the numerically calculated probability boundaries, particularly for the smaller peak freshwater overshoots. However, for larger overshoots, discrepancies arise between

the numerics and theory. At the 1% level the theory overestimates the critical boundary, caused by the asymmetry of the forcing profile. However, at the 50% level, the theory underestimates the critical boundary. The shrinking region of bistability (i.e. presence of another fold bifurcation) and initialising the simulations in equilibrium provide additional sources of error for small diffusive timescales.

We now consider the scenario that time series data for the AMOC is available – here we use synthetically generated data

with an assumed diffusive timescale of 525 years. Performing Bayesian inference (see Methods for details) starting with the uniform prior (purple) we are able to create a tightly constrained posterior distribution (green) centred close to the assumed diffusive timescale, see Figure 7(a).

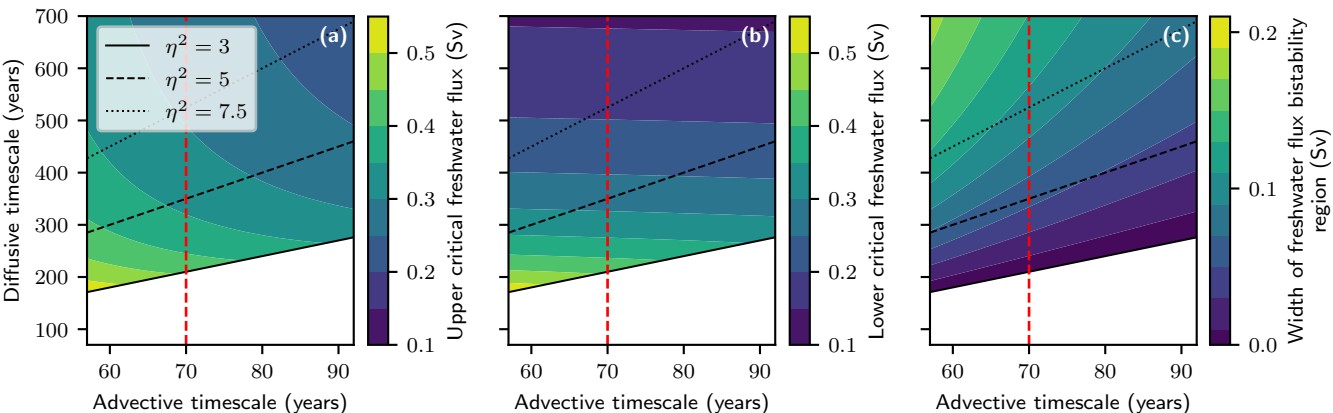

**Figure 5. Critical freshwater fluxes and width of bistability region dependence on advective and diffusive timescales** Colour plots for the location of the critical freshwater fluxes and the bistability region depending on the advective and diffusive timescales. (a) Location of the upper critical freshwater flux that triggers an AMOC collapse from the AMOC on state to the AMOC off state. (b) Location of the lower critical freshwater flux that triggers AMOC recovery from the off to the on state. (c) Width of bistability region, defined by the difference between the upper and lower critical freshwater fluxes. Region plotted corresponds to plausible advective and diffusive timescales as identified by Wood et al. (2019). Black lines are contours of constant ratio between advective and diffusive timescales. Red dashed line denotes the value the advective timescale is fixed at for analysis of an uncertain diffusive timescale.



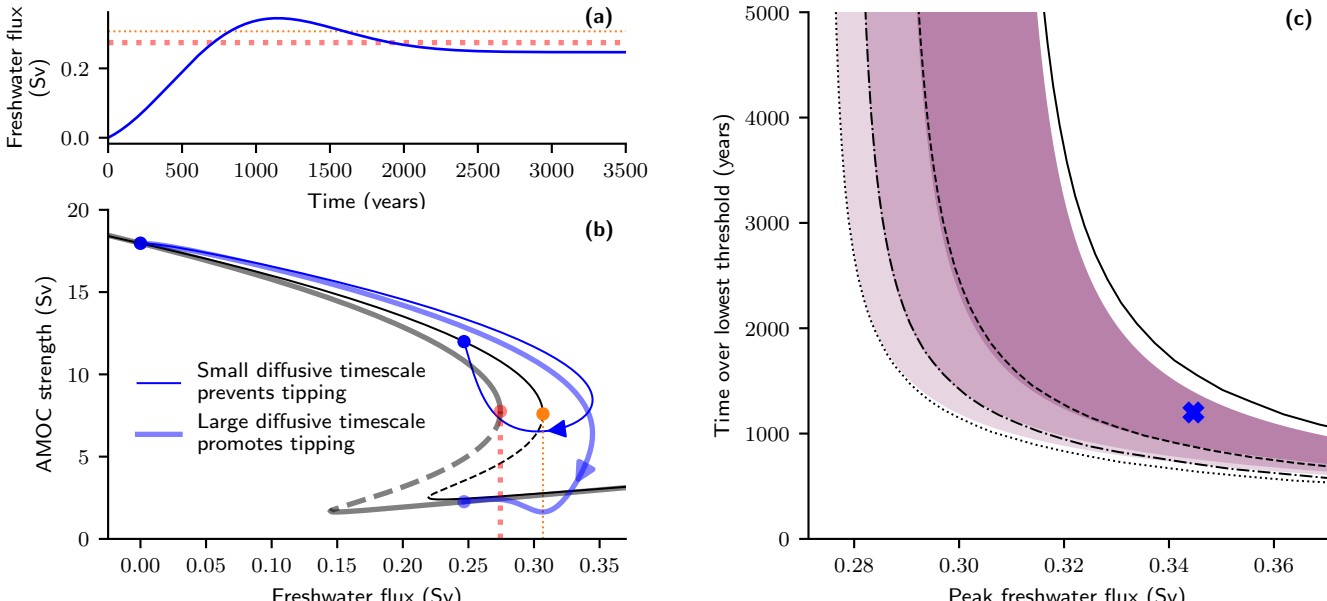

**Figure 6. Probabilistic overshoots given uncertainty in diffusive timescale of AMOC Stommel-Cessi model.** (a) Time profile of freshwater flux that briefly overshoots before stabilising at just below 0.25 $Sv$ ($1Sv = 10^6 m^3/s$). Tipping thresholds are given by the red (low threshold) and orange (high threshold) dotted lines for large and small diffusive timescales respectively. (b) System responses (blue) subjected to the freshwater flux profile given in (a). System with high threshold (thin and opaque) avoids, whereas, the same system but with an early threshold (thick and translucent curves) undergoes tipping. Steady states indicated by black curves, are either stable (solid) or unstable (dashed). Orange and red dots indicate threshold location (fold bifurcation) of the respective systems. (c) Probabilistic critical boundaries separating overshoots that avoid tipping from those that do not, in the plane of time over the lowest possible threshold and peak forcing amplitude, given a uniform distribution in the restoring force proportionality factor. Purple colour gradient shows different probability mass levels derived from the theory. Specifically, from left to right: the start of the lightest purple region indicates the location of the 1% probability of tipping critical boundary; the medium shade 10%; and the darkest shade starts at 25%; and finishes at 50%. Black curves provide the exact boundaries calculated numerically (dotted – 1%, dash-dot – 10%, dashed – 25%, solid – 50%). Blue cross corresponds to time profile of freshwater flux given in (a).

The tightly constrained posterior results in a dramatic reduction in the uncertainty of the mitigation window, see comparison of purple to green in Figure 7(b). Previously, overshoots that had a 25% probability of tipping can be classified as *very unlikely*
with less than 1% probability of tipping. Furthermore, as can be inferred from Figure 5(b), the stabilisation level is within the bistability region (more than 99% confidence). Note that the lower critical freshwater flux threshold is only below the stabilisation level for diffusive timescales greater than 400 years. Thus, if the reversal in freshwater flux is too slow for a given peak freshwater flux, tipping will occur instead of observing a guaranteed recovery. This ultimately means that, where previously the probability of tipping was 50%, now with the constrained posterior the probability of tipping is greater than
99% (not shown).





Whereas the analysis in Figure 7(b) covers a whole spectrum of different overshoot trajectories, Figure 8 performs a more in-depth analysis of how the probability of tipping for a single overshoot trajectory changes based on the distribution of the diffusive timescale. A zoomed in view of the overshoot trajectory is given in Figure 8(a). For this particular overshoot there is a critical diffusive timescale such that smaller diffusive timescales will prevent the AMOC from collapsing. The tipping threshold

that corresponds to this critical diffusive timescale is plotted in black. If the diffusive timescale is smaller, the tipping threshold will be higher, meaning that the overshoot of the threshold will be smaller and for a shorter duration (compare orange line with black line). Let us now consider some uncertainty on the diffusive timescale, centred around a reference value $t_d^{\text{ref}} = 475$ years that corresponds to the orange threshold $p_b^{\text{ref}} = 0.303$ Sv. The orange banding represents the threshold locations $p_b$ that arise from a nonlinear transformation of $t_d$ within one standard deviation (125 years) of $t_d^{\text{ref}}$. Note that the nonlinear relation between

the diffusive timescale parameter and the threshold location makes the orange band asymmetric. Despite an assumed normal distribution on the diffusive timescale the distribution of thresholds is not normally distributed. Visibly, within one standard deviation of the mean includes both timescales above the critical level that would cause the AMOC tip, and timescales that would avoid the system crossing the threshold altogether for the same overshoot trajectory.

In Figure 8(b) the probability of tipping is plotted based on the mean and standard deviation of the normally distributed

diffusive timescale. The orange cross corresponds to the mean and standard deviation given in Figure 8(a). If the standard deviation of the distribution is zero (i.e. the diffusive timescale is known), then, without stochastic variability in the system,

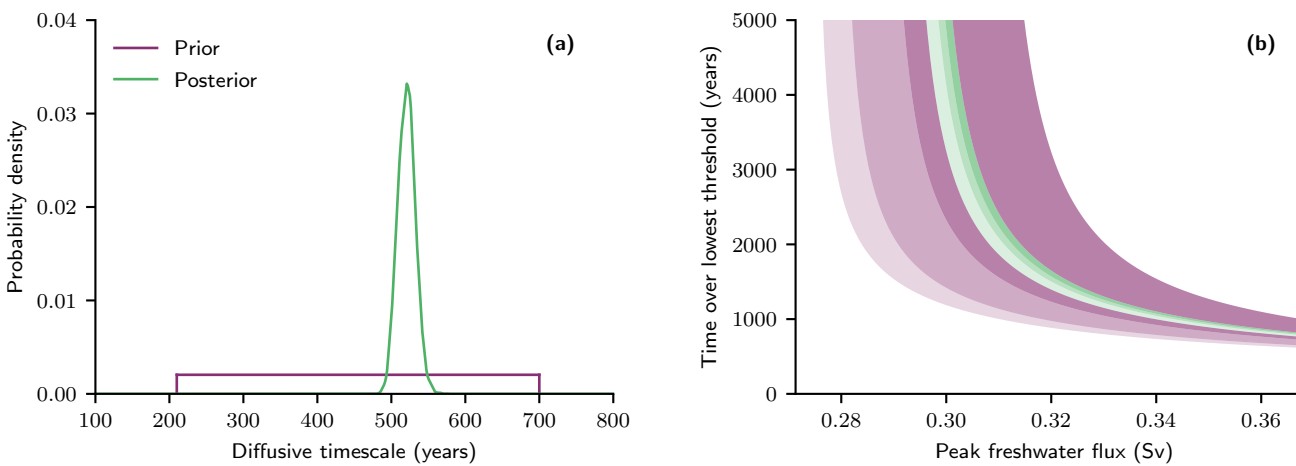

**Figure 7. Constraining uncertainty in diffusive timescale minimises uncertainty in overshoot boundary separating tipping and avoiding tipping.** (a) Probability density functions for diffusive timescale. A uniform distribution is used as a prior (purple), whereas, the posterior has been calculated by performing Bayesian inference on synthetic data generated for an assumed diffusive timescale of 525 years (green). (b) The theoretical probabilistic critical boundaries separating overshoots that avoid tipping from those that do not, in the plane of time over the lowest possible threshold (given the prior distribution) and peak freshwater flux amplitude, are given in colour corresponding to the distributions given in (a). Specifically, from left to right: the start of the lightest shading indicates the location of the 1% probability of tipping critical boundary; the medium shade 10%; and the darkest shade starts at 25%; and finishes at 50%.





the probability of tipping is either zero or 100%. Increasing the standard deviation a little will create some uncertainty close to the critical diffusive timescale for the specific trajectory, but still for most of the cross-section the probability of tipping will be close to zero or 100%. As the standard deviation increases, more distributions will include the critical diffusive timescale with

some non-small probability and therefore create greater uncertainty. Thus, the region of uncertainty spreads out for increasing standard deviation as shown numerically by the colouring.

Alternatively, we can make use of the theory to derive the probability of tipping given the characteristics of the distribution and the overshoot trajectory. The inverse square law, (1), can be written purely in terms of the diffusive timescale and the

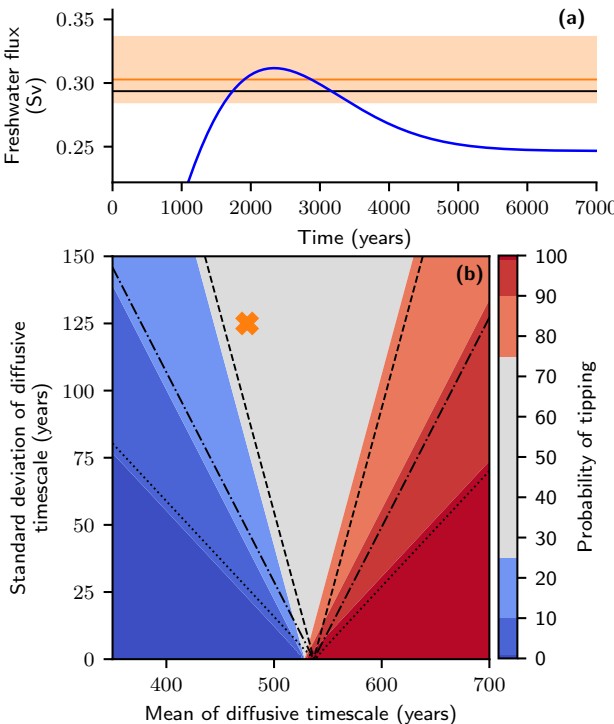

**Figure 8. Probability of avoiding tipping for a single overshoot trajectory based on diffusive timescale distribution** (a) Time series of overshoot trajectory (zoomed in) that starts with zero freshwater flux, reaches a maximum level before declining and stabilising at a freshwater flux of just below 0.25 $Sv$. The black horizontal line provides the location of the tipping threshold at the critical diffusive timescale separating tipping from not tipping. The orange line (mean) and banding (mean $\pm 1$ standard deviation) show the location of the critical threshold for a diffusive timescale parameter distribution that is normal with mean 475 years and standard deviation 125 years (denoted by orange cross in (b)). (b) Plot of the probability of tipping for the overshoot given in (a) depending on the characteristics of a normally distributed diffusive timescale parameter. Colour gives sampling based, numerically calculated probability of tipping and black contours the theoretical probability of tipping derived from inverse square law relationship.





properties of the overshoot trajectory:

$$p_{peak} - p_b(t_d) - \sqrt{-\frac{\ddot{p}(t_{peak})}{2\kappa(t_d)}} < 0, \tag{2}$$

where $\ddot{p}(t_{peak})$ is the curvature of the overshoot trajectory at the peak forcing. The left-hand side is a monotonically increasing function for increasing diffusive timescale, such that there exists a single critical timescale that, for a given overshoot trajectory, separates tipping from not tipping. Specifically, all diffusive timescales less than the critical timescale avoid tipping, meaning that the corresponding cumulative distribution evaluated at this critical value provides the probability of tipping. The theoretical

contours are added to Figure 8(b) as black lines of different line styles and show a good agreement to the numerically calculated probability given by the colour plot. The discrepancy arises from determining the value of the critical diffusive timescale. This is best identified on the x-axis for zero standard deviation, where the black contours converge roughly 10 years larger than the convergence of the colour.

Figure 8(b) shows that if the standard deviation is reduced, but the mean of the diffusive timescale kept fixed (i.e. move

down vertically), then the uncertainty in the probability of tipping reduces. However, generally when reducing the uncertainty (standard deviation), the mean will also likely change. In some scenarios, it is possible for the uncertainty in the probability of tipping to increase despite a reduced uncertainty in the timescale. For example, the initial distribution, might have a large standard deviation but a mean (or mode) far from the critical timescale. If, when reducing the standard deviation, the mean moves closer to the critical timescale, then, it is possible for the probability of tipping to become more uncertain (move closer

to 50%). Importantly, this is just for the single pre-defined overshoot trajectory, but if instead more (or all possible) overshoot trajectories are considered then the overall uncertainty in the probability of tipping will still reduce.

## 3 Conclusions

In this paper, we studied how uncertainty on model parameters can propagate to uncertainty in the probability of tipping for systems subjected to overshoot trajectories. The location of the threshold and the linear restoring rate are two key characteristics

that were identified and individually isolated to examine their importance for the possibility of avoiding tipping. The location of the threshold was the characteristic found to have the most influence. For a given overshoot trajectory, the threshold location will simply determine if an overshoot of the threshold occurs. Assuming an overshoot of the threshold does occur, if the threshold was further away, both the peak overshoot distance and the time spent over the threshold would be reduced. These properties of the overshoot feature in the left-hand side of the inverse square law, (1), while the right-hand side, interpreted as

an upper bound for the mitigation window, remains fixed given a fixed linear restoring rate. Minimising the peak overshoot distance and especially the time over, reduces the likelihood of exceeding the upper bound of the mitigation window, and therefore avoiding tipping.

Another source of uncertainty can be in the linear restoring force to the stable equilibrium, which propagates into uncertainty in the upper bound for the mitigation window (the right-hand side of (1)). The linear restoring force features in the denominator

of the right-hand side, and therefore a weaker restoring force will help prevent tipping by increasing the upper bound for the




mitigation window. This can be intuitively understood by a weaker restoring force causing the system to further lag the stable quasi-static state under a change in external forcing. This acts as building a bigger buffer for when the system crosses the threshold, that causes any tipping to be further delayed and therefore present additional opportunity to reverse the forcing and avoid tipping.

We utilised a simple model for the Atlantic Meridional Overturning Circulation (AMOC) to demonstrate how uncertainty in the diffusive time scale parameter propagates simultaneously to uncertainty in the location of the threshold and the linear restoring rate. Although the advective timescale is well constrained across climate models, a large uncertainty remains in the diffusive timescale. This ultimately results in uncertainty in the probability of tipping for overshoot scenarios. For the AMOC, this translates to a large uncertainty in the probability of the AMOC collapsing.

However, suppose the parameter uncertainty can be constrained, for instance by performing Bayesian inference on observational data. As we have seen for our synthetically generated data, this can greatly reduce the uncertainty in the probability of tipping. For a parameter distribution with a known cumulative distribution function, the probability of tipping can be efficiently calculated in terms of computational costs by using the inverse square law relationship. This avoids a sampling based approach. However, further research is required to understand how the theory can be applied to multiple uncertain parameters in more
complex models.

Some conclusions can be drawn towards decisions about alternative mitigation pathways. To this end, we define an overshoot budget as the product of the peak overshoot distance and time duration of the threshold exceedance. Then, for a fixed overshoot budget, the inverse square law relation (1) suggests that mitigation pathways with large but short overshoots are preferable to small and long overshoots. However, uncertainties in the threshold location can reverse these conclusions for mitigation path-
ways. For instance, small and long overshoots may no longer involve an overshoot of the threshold at all given the uncertainty in the threshold location.

Considering the added possibilities of tipping via other mechanisms can further change the conclusions. For instance, if variability is considered (i.e. with the added possibility of noise-induced tipping), the longer a system spends close to, or beyond the threshold, the easier it is for a system to be triggered into tipping Ritchie et al. (2019). Therefore favouring large
and short overshoots as opposed to small and long overshoots. Furthermore, if rate-induced tipping was possible, then multiple critical rates can arise for the same peak external forcing Ritchie et al. (2023) and therefore constraining system uncertainties becomes even more critical.

Alternative profile shapes may also reduce the distinction between large but short and small but long overshoots Enache et al. (2024). Moreover, uncertainties in the overshoot profile characteristics also need to be considered. It is unlikely that any
overshoot will be performed as planned, both in terms of the peak overshoot distance and time spent over the tipping threshold.

For a simple conceptual model of the AMOC, we find that for any sized overshoot, provided the duration is less than 800 years, tipping would very likely be avoided. Importantly, this encompasses most policy-relevant overshoots under consideration (see e.g. Kikstra et al. (2022)), and therefore suggests AMOC tipping is very unlikely. However, it is important to note that these timescales are likely to be much longer than those observed in climate models (or the real world) Jackson et al. (2022).





Box models for the AMOC, such as that used for this study, tend to omit important advective responses that would otherwise make the response time faster Jackson and Wood (2018b).

Current rates of anthropogenic emissions make crossing climate tipping thresholds increasing likely, despite not knowing their exact location. This study highlights the influence the threshold location and linear restoring rate has on the uncertainty in the probability of tipping. Therefore, constraining the uncertainty in these characteristics is crucial if we want to avoid elements

tipping under overshoot scenarios.

## Appendix A: Methods

### A1 Forcing profiles

Two types of overshoot profile are used in our analysis. The first forcing profile used is a symmetric overshoot given by

$$p(rt) = \Delta_p \operatorname{sech}^2(r(t - t_{peak})), \tag{A1}$$

that starts and finishes at zero, has an amplitude $\Delta_p$ and the rate of change is controlled by $r$.

The second type of overshoot trajectory takes the form of a more realistic profile, first introduced by Huntingford et al. (2017):

$$p(t) = p_0 + \gamma t - (1 - \exp(-\mu(t)t))(\gamma t - (p_{\lim} - p_0)). \tag{A2}$$

This forcing profile provides the flexibility to start ($p_0$) and finish ($p_{\lim}$) at different levels. The transition between these two

levels is determined by $\mu(t) = \mu_0 + \mu_1 t$, where $\mu_0$ and $\mu_1$ determine the maximum amplitude and time to converge to the finishing level. The parameter $\gamma = r - \mu_0(p_{\lim} - p_0)$ is chosen to ensure that all profiles have the same initial rate of increase, determined by $r$.

### A2 Overshoot theory for arbitrary threshold

The overshoot theory, as given by (1), details the time allowed over the tipping threshold. However, if the threshold location is

uncertain, we would like to generalise this to the time over an arbitrary threshold, $p_{thr}$ – in our case we use the earliest tipping threshold according to the initial distribution. We follow a similar approach used in the original derivation in Ritchie et al. (2019), starting with the overshoot theory given by:

$$p_{peak} - p_b < \frac{1}{a_0} \sqrt{-\frac{\ddot{p}(t_{peak})}{2\kappa}}. \tag{A3}$$

The Taylor expansion of the forcing profile, $p(t)$, about the peak level of forcing $p_{peak}$ at time $t_{peak}$, is given by



$$p(t) \approx p_{peak} + \frac{1}{2}\ddot{p}(t_{peak})(t - t_{peak})^2. \tag{A4}$$

Using (A4), let us consider the time over, $t_{over}$, an arbitrary threshold, $p_{thr} < p_{peak}$:

$$t_{over} = 2\sqrt{\frac{2(p_{peak} - p_{thr})}{-\ddot{p}(t_{peak})}}. \tag{A5}$$

Rearranging (A5) for $\ddot{p}(t_{peak})$ and substituting into (A3) gives the expression for the time allowed over an arbitrary threshold

$$t_{over}^2 < \frac{4(p_{peak} - p_{thr})}{a_0^2 \kappa (p_{peak} - p_b)^2}. \tag{A6}$$

Importantly, if the threshold is chosen to be the tipping threshold ($p_{thr} = p_b$) then (A6) reduces to the original inverse square law, (1).

### A3 Prototypical fold bifurcation model

A conceptual model is used to study how uncertainty in either the location of the tipping threshold or the linear restoring force propagates to uncertainty in the probability of tipping for the overshoot scenarios. The model described by

$$\dot{x} = a_0 \left( p_b - p(rt) - \kappa(x - \epsilon)^2 \right), \tag{A7}$$

allows the threshold location, $p_b$, and the linear restoring force, proportional to $\kappa$, to be individually isolated. For simplicity, the inverse timescale parameter $a_0$ is set to one. The parameter $\epsilon = x_0 - \sqrt{p_b/\kappa}$ ensures that the system is initialised at the same starting position, $x_0 = 2.5$, without changing the threshold location or the linear restoring force relative to the distance to the threshold.

Figures 1–4 are created using the conceptual model (A7) coupled with the symmetric overshoot forcing (A1). The parameters used to generate the forcing in Figure 1(a) are $\Delta_p = 2.35$, $r = 0.24$, and $t_{peak} = 25$. This is then applied, in Figure 1(b) to the conceptual model, with parameter value $\kappa = 1$, and for the thick, translucent curves, $p_b = 2$, while for the thin, opaque curves $p_b = 2.3$. The probability of tipping is calculated based on a uniform distribution for the threshold location, $p_b \sim \mathcal{U}[2.0, 2.3]$, in Figure 1(c). For each overshoot trajectory there is a critical threshold location. On the one hand, there is the numerical probability of tipping, which is calculated using a bisection method and indicated by the black lines ranging from dotted to solid. Note, a given probability level corresponds to a unique critical threshold location determined from the probability distribution. For this given location, a bisection method is applied to find the critical time duration over an arbitrary threshold (we choose the lowest threshold in the distribution) for each peak external forcing level.

On the other hand, there is the theoretical approach using the inverse-square law, (1) or (A6). Similarly, the critical threshold location for a given probability level is determined from the cumulative probability distribution function (either analytically





or numerically). Then the inverse square law can be directly applied to determine the exceedance time for any given peak forcing level. A similar philosophy and approach is used for all probability of tipping figures. For Figure 2, the same uniform distribution is used as in Figure 1, and the knowledge-based distribution is normally distributed, $p_b \sim \mathcal{N}(2.1, 0.02)$.

In contrast, Figures 3–4, keep the threshold location fixed, $p_b = 2$, and vary the linear restoring force proportionality constant,
$\kappa$. The specific overshoot trajectory, in Figure 3(a), is created using the parameters, $\Delta_p = 2.1$, $r = 0.1$, and $t_{peak} = 40$. For Figure 3(b), the thin, opaque curves correspond to $\kappa = 1$ and the thick, translucent curves, $\kappa = 2$. Figure 3(c) assumes a uniform distribution for the linear restoring force, $\kappa \sim \mathcal{U}[0.25, 3.25]$, which is also used as the initial distribution in Figure 4. The alternative distribution also uses a uniform distribution for the linear restoring rate, $\kappa \sim \mathcal{U}[3.5, 6.5]$, while the knowledge-based distribution is normally distributed, $\kappa \sim \mathcal{N}(1, 0.25)$.

## A4 Stommel-Cessi AMOC model

We use a model for the AMOC introduced by Cessi (1994), which is a modification of the 2-box Stommel model Stommel (1961), and describes the change in non-dimensional salinity flux, $x$:

$$x' = \tilde{p}(t) - x(1 + \eta^2(1 - x)^2). \tag{A8}$$

The non-dimensional freshwater flux is given by $\tilde{p}(t)$, and the single parameter, $\eta^2 = t_d/t_a$, defines the ratio of the diffusive,
$t_d$, to advective, $t_a$ timescales. The fold bifurcation points, $(\tilde{p}_\pm, x_\pm)$, for (A8), are given by Lux et al. (2022):

$$\tilde{p}_\pm = \frac{2}{3} + \frac{2}{27}\eta^2 \mp A\left(\frac{2}{9}\eta^2 - \frac{2}{3}\right), \qquad x_\pm = \frac{2}{3} \pm A, \tag{A9}$$

where

$$A = \sqrt{\frac{1}{9} - \frac{1}{3\eta^2}}.$$

In our analysis, we keep the advective timescale fixed ($t_a = 70$ years), but the uncertainty in the diffusive timescale, $t_d$,
is much larger (plausible values range from 70 to 700 years) and so $t_d$ is considered uncertain in our study. In the original formulation of the Stommel-Cessi model, (A8), the scaling between dimensional and non-dimensional time, and freshwater flux depends on the diffusive timescale. Therefore, under scaling to non-dimensional units the same non-dimensional freshwater flux profile will generate different dimensional flux profiles for different diffusive timescales. Instead, we rescale with respect to the advective timescale, which changes (A8) to

$$\dot{x} = p(t) - \frac{x}{t_d}(t_a + t_d(1 - x)^2), \tag{A10}$$

where the scalings to non-dimensional quantities are given by





$$\Delta S = \frac{\alpha_T \theta}{\alpha_S} x, \quad t = t_a \tau, \quad F(t) = \frac{\alpha_T \theta H}{\alpha_S S_0 t_a} p(t) = \xi p(t). \tag{A11}$$

A description of the parameters and their values can be found in Table A1. The dimensional AMOC flow strength, $Q(x, t_d, V)$, measured in Sverdrups $(S_v)$ is given by:

$$Q(x, t_d, V) = \frac{\gamma V}{\beta t_a t_d}(t_a + t_d(1 - x)^2), \tag{A12}$$

where $V$ is the ocean volume.

For (A10), the non-dimensional fold bifurcation points for $x_\pm$ remain unchanged, as given by (A9). It can be shown that the non-dimensional freshwater flux fold points get adjusted by $p_\pm = \tilde{p}_\pm/\eta^2$. The dimensional freshwater flux fold points are then scaled according to (A11) to generate Figure 5.

Figure 6(a) is created using the overshoot profile given by (A2) and scaled according to (A11), with parameter values: $p_0 = 0$, $p_{\lim} = 1/\xi$, $\mu_0 = -0.05$, $\mu_1 = 0.0057$ and $r = 0.01$. In Figure 6(b), the overshoot profile is applied to system (A10), with $t_d = 455$ for the thin, opaque curves and $t_d = 700$ for the thick, translucent curves. The AMOC flow strength has been calculated using (A12), where the ocean volume, $V$, is chosen such that the initial AMOC strength is equal to $Q_0 = Q(0, 525, V_0)$. The reference volume $V_0$, given in Table A1, is an approximate value for the ocean volume based on General Circulation Models
Wood et al. (2019). A diffusive timescale of $t_d = 525$, corresponds to $\eta^2 = 7.5$ as used in the literature Cessi (1994). The probability of tipping in Figure 6(c) is calculated based on a uniform distribution for the diffusive timescale, $t_d \sim \mathcal{U}[210, 700]$. The same distribution is used as the prior in Figure 7 and where the posterior is calculated using a Bayesian Inference procedure as described in the following section.

Figure 8(a) is created using the overshoot profile given by (A2) and scaled according to (A11), with parameter values:
$p_0 = 0$, $p_{\lim} = 1/\xi$, $\mu_0 = 0.005$, $\mu_1 = 0.0009$ and $r = 0.01$. For this overshoot profile, there will be a single critical diffusive

**Table A1.** Description of parameters and their values used in the Stommel-Cessi model

| Parameter | Description | Value (Units) |
|---|---|---|
| $t_a$ | Adevective timescale | $70\ (yrs)$ |
| $\alpha_T$ | Thermal expansion coefficient | $10^{-4}\ (K^{-1})$ |
| $\alpha_S$ | Haline contraction coefficient | $7.6 \times 10^{-4}\ (psu^{-1})$ |
| $\theta$ | Meridional temperature difference | $25\ (K)$ |
| $H$ | Mean ocean depth | $4{,}500\ (m)$ |
| $S_0$ | Reference salinity | $35\ (psu)$ |
| $\beta$ | Seconds in a year | $3.1536 \times 10^7\ (s\ yr^{-1})$ |
| $\gamma$ | $m^3\ s^{-1}$ to $Sv$ conversion | $10^{-6}\ (Sv\ s\ m^{-3})$ |
| $V_0$ | Reference volume | $3.5 \times 10^{16}\ (m^3)$ |

timescale that separates tipping from not tipping. The critical timescale can either be calculated numerically or theoretically from (2), both using a bisection method. The corresponding tipping threshold can then be calculated using (A9) together with the scalings in (A11). The black horizontal line in Figure 8(a) corresponds to the threshold for the numerically calculated critical timescale. Similarly, the orange line provides the threshold for $t_d = 475$ and the orange banding the thresholds that lie

within diffusive timescales, $t_d \in [350, 600]$. This interval corresponds to $\pm 1$ standard deviation of a normal distribution with mean $475$ and standard deviation $125$, which corresponds to the orange cross in Figure 8(b). The probability of tipping is determined by evaluating the cumulative density function at the critical timescale (numerically for colour and theoretically for black contours) using the respective mean and standard deviation of the normal distribution.

## A1    Bayesian Inference

If time series data for the AMOC is available, this can be used to constrain the uncertainty in the model parameter. This ultimately leads to a tighter constraint on the critical boundary separating tipping scenarios from those where tipping can be prevented (for a visualisation of probabilistic critical boundaries, see Figure 7b). Here, we use a Bayesian inference approach Stuart (2010). The procedure is analogous to the one used in Lux et al. (2022), especially in terms of the discrepancy model, the likelihood function, and the generation of the synthetic time series. The assumed true value for $t_d$ in our dimensional setting is

$525$ years and the underlying ODE is given by (A10). We use a Markov chain Monte Carlo (MCMC) approach Brooks et al. (2011), where the idea is to obtain the desired data-informed (posterior) distribution as the invariant distribution of the Markov chain over the prior support.

The prior distribution for $t_d$ is assumed to be uniform, and based on the assumption that the ratio, $\eta^2$, of the diffusive to advective ($t_a$) timescales is between 3 (minimum ratio for the existence of multistability of the AMOC) and 10, where $t_a$ is

fixed to 70 years, leading to $t_d$ being uniformly distributed over $[210, 700]$. We obtain the posterior (data-informed) distribution by running a MCMC algorithm provided in the MATLAB-based software framework UQLab Marelli and Sudret (July 13-16, 2014), version 1.3.0. using the affine invariant ensemble sampler with $100$ Markov chains with $400$ steps (see the manual Wagner et al. (2019) for a detailed documentation).

*Author contributions.* K.L-G. and P.D.L.R. contributed to the design of the study and wrote the manuscript. K.L-G. performed the Bayesian
inference and P.D.L.R. carried out the numerical analysis.

*Competing interests.* There are no competing interests

*Acknowledgements.* K.L.-G. acknowledges support of the EU within the TiPES project funded by the European Union's Horizon 2020 research and innovation programme under grant agreement No. 820970., and funding by the Irène Curie fellowship. P.D.L.R. was supported



by the European Research Council "Emergent Constraints on Climate-Land feedbacks in the Earth System (ECCLES)" project, grant agreement number 742472, and by the Optimal High Resolution Earth System Models for Exploring Future Climate Changes (OptimESM) project, grant agreement number 101081193. The authors would like to thank Richard Wood for his valuable comments that helped improving an earlier version.



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
