# Peer review of "Uncertainty quantification for overshoots of tipping thresholds"

_EGUsphere, 2024_

## Referee Comment (RC2)

**Review of manuscript egusphere-2024-2170**

'*Uncertainty quantification for overshoots of tipping thresholds*' by Lux-Gottschalk & Ritchie

**Summary**

The study by Lux-Gottschalk & Ritchie explores how parametric uncertainty introduces and impacts uncertainty in the response of tipping elements to overshoot trajectories, which temporarily exceed the critical tipping threshold.

First, the impact of uncertainties in the locating of the critical threshold and the linear restoring rate on the response of tipping elements to overshoot trajectories of varying overshoot magnitudes and durations is illustrated based on a simple dynamical system with a fold bifurcation (Sect. 2.1 and Sect. 2.2). In particular, the tipping element response is described in terms of probabilities of tipping, based on a modification of a previously derived inverse square law (Ritchie et al. 2019). Therein, the uncertainty in the location of the critical threshold leads to a larger uncertainty in the tipping response to overshoot trajectories than the linear restoring rate. Constraining the uncertain parameter can reduce uncertainties in this tipping response. In addition, the effect of changes in the mean vs the range of the distribution of the uncertain parameter is explored. In a next step (Sect. 2.3), this framework is extended to a box model of the Atlantic Meridional Overturning Circulation with a similar bifurcation structure and an uncertain parameter (diffusive timescale) that concurrently translates into an uncertain location of the critical freshwater flux threshold and the linear restoring rate.

**General comments**

The paper approaches the relevant and scientific interesting question of uncertainty in the tipping response to overshoot trajectories in the presence of uncertain parameters from a theoretical / conceptual side. The results could perhaps serve as a starting point for further (probabilistic) assessments of potential critical transitions in the Earth system.

The introduction contains relevant information on tipping and the concept of 'safe' overshoots. In some places, it seems repetitive or related information is spread across the introduction. Some restructuring of the introduction might help to avoid these repetitions and it could also support making the motivation for and main aims of this study clearer. For example, tipping is introduced and defined in L19-25, and some additional general information is added in L52-54. Impacts of tipping are mentioned in L26, and in L45-46 the potential consequences of AMOC tipping are described. Uncertainties are first referred to in L32-40, and laid out in more detail in L77-82.

While the approach of explaining the basic interplay between the overshoot trajectory and the location of the critical threshold, the linear restoring rate or the diffusive timescale (in the AMOC box model) in the beginning of each section before introducing uncertainty in these parameters is a good starting point for understanding the key figures displaying tipping probabilities, it is not always easy to follow the results presented in this study. This is for two main reasons:

1) The clarity of the language could be improved throughout the manuscript. In some places, the language distracts from the key messages of the paper. These key messages might be formulated more concisely. Please also see the specific comments below.

2) The methods are not given in the main manuscript but described in the Appendix. To allow the reader to properly follow the results, the models and the derivation of the tipping probability based on the (modified) inverse square law and numerical simulations should be described before presenting the impact of uncertainty in the location of the critical threshold and the linear restoring rate on the tipping response. In addition, the Bayesian Inference to constrain uncertain model parameters should be explained in such a section.

The manuscript illustrates how parametric uncertainties translate into uncertainties in the tipping response based on conceptual models, and I acknowledge that a lot can be learnt from these approaches, e.g. to showcase that constraining parametric uncertainty is important to constrain the tipping response to overshoot trajectories. It could be interesting to expand the discussion of if / to what extent / how this framework and the conceptual results could be applied to more complex situations.

I have included more specific comments, questions and suggestions below.

**Specific comments**

L01: Maybe replace 'Earth' by 'Earth system' or 'climate system'. In addition, can these subsystems be considered to currently be in a stable (steady) state?

L03: Maybe replace 'common mechanism of tipping' by 'possible mechanism of tipping'.

L05: Maybe replace 'is possible' by 'could be possible'.

L07: Please give references in brackets here and elsewhere if not integrated into the sentence, i.e. (Ritchie et al., 2019).

L07: What is meant by 'other system features'? Please specify.

L08: I would like to suggest to add 'in response to a temporary overshoot' (or similar) after 'probability of tipping'.

L14-15: This statement appears as a key result / conclusion of the analysis in the abstract. I was wondering whether this is the case, e.g. when reading L166-168, and also what it implies for conclusions on mitigation pathways (as indicated in the previous sentence).

L16-17: To me, the sentence suggests that many parameters in the box model have been tested and the diffusive timescale has been identified as particularly relevant for determining uncertainty in the AMOC tipping behaviour. I would like to suggest to rephrase to: 'In addition, we illustrate how constraining the highly uncertain diffusive timescale within this box model reduces the tipping uncertainty of the AMOC in response to overshoot scenarios' (or similar).

L20: 'sudden' could suggest that tipping occurs abrupt in time, which may not necessarily be the case. Please consider to reformulate.

L22: 'kicked off their current stable state' could refer noise- and rate-induced tipping, but not necessarily bifurcation-induced tipping, which is associated with the loss of stability beyond a critical threshold in some control parameter, followed by a critical transition to a remaining stable state. Please consider to reformulate.

L28: Maybe replace 'are assumed' by 'have been suggested' (or similar).

L29-31: Maybe replace 'can still be avoided' by 'could potentially be avoided'. I would also like to suggest to explicitly acknowledge that this statement is based on model simulations.

L55-72: I am not sure how helpful the description of the different mathematical mechanisms of tipping is, in particular given that the relation to overshoots is not directly described. Please consider to connect the different mechanisms to overshoot trajectories. This may also help to justify the focus on bifurcation-induced tipping in this paper as stated in L73.

L88-89: Please check the section numbering in the Appendix. In addition, I would also like to suggest to move (parts of) the Appendix into a section that focuses on the Methods in the main part of the manuscript (see general comments). This might be helpful for readers that are not familiar with the fold bifurcation and the AMOC box model, and would allow to remove some of rather detailed information of the AMOC box model in the introduction (L86-89). This also applies to the probabilistic extension of Ritchie et al. (2019) and the inverse square law, as its development is described as a main aim of this paper (L90).

L91: Maybe replace 'fold tipping scenario' by 'tipping via a fold bifurcation' (or similar).

L119: Why is the location of the critical threshold the 'biggest uncertainty to consider'? Is this already a result of the analysis, based on the argument that follows this statement, or motivated by previous studies?

L126: Maybe replace 'large' by 'far'.

L130: 'early' could be associated with a time dimension. Please consider changing 'early' to 'low' (or similar).

L131: I am not sure about the formulation 'otherwise identical system'. The bifurcation diagrams also differ in e.g. the location of the basin boundary in addition to the location of the critical threshold.

L133: Maybe replace 'a threshold further away' by 'a higher threshold' (or similar).

L136: I would like to suggest to use a different terminology for cases without tipping. In the manuscript, formulations such as 'tipping can be avoided' are often used when there is no critical transition due to a high tipping threshold (or similar; see e.g. L158-159, L177, L185-186, L307-308, L323). This formulation could be misunderstood.

L137-138: I am not sure whether it is clear how a tipping probability is assigned here. Please see the related comments on including a section on the methods in the main part of the paper.

L139: While it may be clear to some readers, it might still be helpful to add a short explanation for the choice of this range for the location of the critical threshold.

L140: I would like to suggest to indicate the quantities on the axis of Figure 1(c) (that is, the time spent over the lowest possible threshold and the peak in external forcing) in Figure 1(a). In addition, maybe replace 'based on' by 'depending on'.

L142-149: Please indicate the probability ranges (in terms of the shading and the black lines) in Figure 1(c) as an additional legend, as in Figure 8(b).

Figure 1(c): Is there a reason why the colouring stops at tipping probabilities of 50% and larger, in contrast to Figure 8(b) and given that for a known threshold (without uncertainties) there would only be one line separating *tipping* (100% probability of tipping) from *non-tipping* (see also e.g. L310-312), if I understand correctly.

L150-155: It could be helpful to describe more explicitly in the beginning of the description of the results how *uncertainty* in the response to overshoot trajectories and a low / high *tipping*

*probability* relate. Only at the end of Sect. 2 (L334-335) a higher uncertainty is linked to a tipping probability moving closer to 50%.

L151: Please consider rephrasing to 'peak forcing not even overshooting the threshold' (or similar) instead of 'profile not even overshooting the threshold'.

L152-153: I am not sure about the structure of this sentence. Please check.

L161: Please specify 'distribution' here and elsewhere (e.g. L209). I assume that in this case it is referred to the 'distribution of the location of the critical threshold'.

L166-168: I was wondering whether it might be possible to align the (description of the) probabilities with the language used in the IPCC.

L171: 'however' could be removed. Maybe replace 'factor' by 'parameter' or 'system characteristic' (or similar).

L171-172: I would suggest to decide for one expression here. In addition, it might be helpful for the reader to give a short explanation of the strength of the linear restoring rate. Is this different or the same for both stable equilibria or is this a general system property (as indicated by the e.g. expressions 'weaker system', 'strong system'; L182-187)? This could also be included in a dedicated section describing the methods in the main part of the manuscript.

L176-177: I am not sure about the formulation 'identical systems', as the systems also differ in e.g. the location of the basin boundary. How does the basin boundary relate to the restoring rate?

L179: Maybe replace 'boundary for the basin of attraction' by 'boundary of the basin of attraction'.

L181: Which stable state do you refer to here (i.e. the upper stable state or the lower stable state or both)?

L183-184: What is meant by 'curvature of the fold'? I am not sure whether this becomes clear without a brief explanation.

L184: What does 'earlier' refer to here? How is time involved?

L198: Please consider rephrasing to: 'the tipping uncertainty is substantially smaller' (or similar). Otherwise the sentence could suggest that an uncertain restoring force is effective in 'reducing uncertainties'.

L205: I think the comma after 'tipping' is not needed here.

L206: Please consider rephrasing to: 'An overshoot trajectory that sits...' (or similar).

L208-209: This sentence is unclear to me. What is meant by 'the separation between the 1% and 50% curves'?

L210: Please specify 'uncertainty' here and elsewhere. It should be clear whether it is referred to uncertainty in tipping or uncertainty in e.g. the location of the critical threshold.

L210: Maybe I have misunderstood this sentence, but to me, it looks as if the alternative distribution covers a different range of the restoring force proportionality factor (3.5-6.5) than its initial distribution in Figure 4(a).

L211: What is meant by the 'width of the banding'? Do I understand correctly that this refers to the 'distance' between the 1% and the 50% probability curves? In general, it might be helpful for the reader to introduce and use a clear terminology when describing Figure 4(b) (and similar figures). So far, different notions have been used throughout the manuscript (e.g. uncertainty in tipping behavior, probabilistic critical boundaries, uncertainty in the tipping probability, uncertainty in the probability of an overshoot avoiding tipping).

L212-213: This sentence is not complete. Please rephrase.

L215-217: These sentences are not clear to me. If I understood correctly, a clear boundary that separates tipping from non-tipping exists for a known restoring rate without uncertainties, and would be given by the inverse square law. What does '(uncertainty of) the location of critical boundary (separating tipping from not tipping)' refer to? How can there be 'little change in the width' and a decrease in the 'uncertainty of the location of the critical boundary' at the same time? Please also see the related comment for L211.

L231: Please consider rephrasing to: 'The chosen ranges of the advective and diffusive timescales correspond to...' (or similar).

L235-259: This is an extensive description of critical freshwater fluxes depending on the advective and diffusive timescales. I would suggest to shorten these paragraphs, if possible, and focus on the aspects that are directly relevant for the overshoot scenarios.

L238: I would like to suggest to add the missing bistability as the main reason (if I understood correctly), which then means that no critical freshwater flux exists.

L257-259: These sentences are unclear to me. What is meant by 'the upper critical threshold moves later'? Does the second sentence refer to the lower critical freshwater flux threshold?

L261: Please add a short justification for a stabilization of the freshwater flux at 0.25 Sv.

L263: I am not sure whether a small diffusive timescale should be referred to as an 'event'. I would like to suggest to rephrase to 'In the case of a small diffusive timescale, the tipping threshold...' (or similar).

L263: 'late' (and 'early') could be associated with time. Please check the formulation here and elsewhere.

L266: Why is the exact number of 0.2 Sv given here? If I understand Figure 6(b) correctly, the AMOC would recover for a reduction in the freshwater flux to below approx. 0.22 Sv. Please also add for which value of the diffusive timescale this threshold is given.

L272: Maybe it could be referred to the figure that shows the prior distribution of the diffusive timescale.

L273: Maybe replace 'recovery of the AMOC on state' by 'recovery to the AMOC on state'.

L278-283: What is meant by 'initialising the simulations in equilibrium' in this context? I think some more explanation may be helpful for the reader here to understand how this acts as an additional 'source of error'. Does 'error' refer to the discrepancy between numerics and theory? Please specify. In addition, how is the applicability of the framework presented here to more complex models effected by the discrepancy between numerics and theory?

L288: Maybe change to 'substantial' or 'strong' instead of 'dramatic'.

L289-290: Does 'previously' refer to the prior distribution? This sentence is not clear to me. This also applies to 'previously' (and 'now') in L294.

L290: How is the 'stabilisation level' defined? I assume that it refers to the forcing level after the overshoot, but a clear definition of this term is missing (or maybe I have missed it).

L293: I am not sure if 'guaranteed' fits well here.

L294-295: It might be helpful to show the >99% probability of tipping (see previous related comment on the probability levels that are shown in the figures).

L332: The comma after 'distribution' may not be needed.

L332-335: If possible, it might be helpful to indicate this example in Figure 8(b).

L335-336: Can this also be inferred from a specific figure?

L338: Maybe replace uncertainty on model parameters' by 'uncertainty of/in model parameters'.

L340-354: In the summary of the role of the location of the critical threshold and the linear restoring rate for the system response to overshoot trajectories, the impact of uncertainties in these parameters on the tipping response could be addressed in more detail (as this is the focus of the paper).

L343: Maybe reformulate and replace 'further away' by 'high' (or similar).

L346: Something may be missing after 'especially the time over'. Please check.

L352: Maybe I misunderstood something, but I got confused by 'quasi-static'. Do the black lines in e.g. Figure 6(b) correspond to equilibrium states or quasi-equilibrium states?

L360-362: I would suggest to combine these two sentences into one sentence, e.g. 'Constraining parameter uncertainty, for instance by performing Bayesian inference on observational data can greatly reduce...' (or similar).

L362-363: Here, it could be helpful to discuss the applicability of the (probabilistic) inverse square law relationship in more complex models (i.e. beyond the conceptual models used here) in more detail.

L366-367: While the concept is interesting, I am not sure whether the introduction of an 'overshoot budget' is needed here. It is not used beyond this paragraph, and even in this paragraph it is not entirely clear to me how it relates to the main part of the manuscript.

L374-375: This sentence is not complete. Please also consider putting this statement (and a similar statement in L367-369) into context – it might still be favourable to avoid overshooting critical thresholds at all, if possible, instead of 'favouring large and short overshoots as opposed to small and long overshoots'.

L375-377: Is the need to constrain system uncertainties in the case of rate-induced tipping specifically related to overshoot scenarios or does it apply in general?

L381: I am not sure whether a 'planned overshoot' fits well as a formulation. Please consider rephrasing.

L383: 'AMOC tipping is very unlikely' is a strong statement, based on the AMOC box model. While limitations are discussed in the following sentences, I would still like to suggest to reformulate and clearly indicate that this statement refers to the box model.

L389: Maybe add 'in response to possible overshoot trajectories' at the end of the sentence.

L389-390: 'elements tipping'?

---

## Author Response (AR1)

**Uncertainty quantification for overshoots of tipping thresholds**
**Reviewer Responses**

We are grateful for the constructive reviewer comments received on our manuscript. These comments are repeated in black, and our responses given in blue.

**Response to Reviewer 1**

In the manuscript "Uncertainty quantification for overshoots of tipping thresholds", the authors investigate how uncertainty on the parameters of a dynamical system yielding a saddle-node bifurcation (namely the tipping threshold and restoring force) leads to uncertainty in the outcome of overshoot scenarios, in which the forcing is higher than the tipping threshold for a finite time.

Notably, the results draw upon the previously established inverse-square law. These are sound and certainly publishable, fitting well the scope of Earth System Dynamics. I however have a series of comments (labelled c#) that I would like to have considered before recommending publication of the manuscript (along with some more taste-dependant suggestions labelled s# at the end)

c1: The way that probabilities are computed from the inverse square law is not announced early enough in the manuscript. If I understand, the method is explained by the prior to last paragraph in Appendix A3. I may have missed it, but the reader is never directed to appendix A3 when mentioning "theoretical probabilistic boundaries" or "boundary levels derived from the theory". This especially brings confusion in the end of section 2.3 from line 317 when reading "we can make use of the theory to derive the probability of tipping", this time via a different method. I would suggest to have this part of appendix A3 in the main text, for example the first time that the method is used.

We have moved the text from the appendix to the main text to before introducing Figure 1(c). We now explicitly state that "For any given overshoot profile, there will be a critical threshold location that separates tipping (lower thresholds) from not tipping (higher thresholds). Therefore, the cumulative probability density function at this critical threshold gives the probability of tipping. This probability of tipping, or more precisely the critical tipping threshold location, can either be calculated numerically (using a bisection method) or via a modification to the inverse square law." Here, we have additionally moved the modified theory that accounts for time over an arbitrary threshold to the main text and referred to appendix A1 for the full derivation as to not disrupt the flow too much.

c2: As much as possible, I would find it clearer if "prior distribution" and "posterior distribution" were systematically used, rather than "prior" and "posterior" alone.

We have now added the word "distribution" at the occurrences of "prior" and "posterior".

c3: L19 - I find the first sentence weird as an introductory statement for this manuscript. Also, I see that climate tipping points are indeed mentioned in the "17 sustainable development goals" but, as written now, it reads as if tipping points themselves are a development goal.

We have now removed the first sentence.

c4: L28 - "Assumed" is incorrect as it is no assumption but some approximation. I believe "thought" would fit better.

We have changed this to read "...have been suggested..."

c5: L73-74 - As far as I can understand, the focus on bifurcation-induced tipping in this study is total. How is the previous paragraph on different tipping mechanisms related to the study? If it is because safe overshoot phenomena are inherently a rate-induced effect, it should be clearer. I believe that this paragraph could be reformulated to motivate that rate-induced effects can also impact tipping behaviours, with safe-overshoot being an example.

The reason for elaborating on different tipping mechanisms is that we would like to emphasize in the introduction that also other mechanisms might cause the system to tip and influence whether we face a "safe overshoot". The focus in our manuscript is indeed on bifurcation-induced tipping. We use the wording "without tipping", meaning that no bifurcation-induced tipping occurs. We make that point in saying "Although all three mechanisms can contribute to the uncertainty in mitigation windows, here, our primary focus is on better constraining the mitigation window for bifurcation-induced tipping.". We agree that the extent to which other tipping mechanisms are introduced should be reduced. Therefore, we have shortened the elaborations on the other tipping mechanisms and moved the discussion on the interplay to the Conclusions section.

c6: L96 - Is it true that this parameter is always the inverse timescale? For the simple systems used I can see it, as $1/a0$ multiplies the left hand side. However, for more complicated systems it is less clear, depending on the definition of the timescale of a system. Maybe "related to the inverse timescale" would be more correct?

We have modified the sentence as suggested. Further, we now provide the precise definitions for $a_0$ and $\kappa$ for a 1d system to help contextualise the inverse square law. We have replaced the $a_0$ in the equation for the conceptual model by a $\tau = -1/a_0$ to more clearly represent the timescale of the system, but keep the $\kappa$ as this precisely agrees with the definition for $\kappa$ in the inverse square law.

c7: L111 - Here and in general, it would be nice to refer to the appendix when appendix content is involved (and also, not only to the equations of the appendix) with sentences like "more about that in appendix #". Also true for example for the forcing (A1) used, to which I see no mention in the main text.

We have now moved various parts of the appendix to the relevant sections in the paper including details about the forcing profiles and the prototypical fold bifurcation model. We have done this to improve the readability of the manuscript while being careful not to disrupt the flow of text too much.

c8: Bayesian inference should be appendix A6.

Thank you for spotting this. Though now, as mentioned above, we have moved the details about the Bayesian inference to the main text as it is a key approach that we wish to highlight as a method for reducing parameter uncertainty.

c9: Fig 1.c - It would be nice to mention somewhere in the text that overshoot occurs in the whole plane, and indeed every time such a figure is presented. It is somehow trivial because of the y-axis name, but it can be missed in a first read.

We now write "The probability of tipping is plotted in Figure 1(c) for a range of overshoot forcing profiles characterised by...". However, it is also important to note that not all of those will actually result in an overshoot of the tipping threshold. This was already noted, to explain the large uncertainty in the tipping probability (for long overshoots), but in the introduction to the figure we now also write "Note however, not all of these forcing profiles will result in an overshoot of the tipping threshold, particularly if the threshold is far away."

c10: L176 - I would not describe these systems as identical. They have a different bifurcation diagram, but a saddle-node occurring at the same forcing, and the same stable state at zero forcing. It implies that rate-induced effects are more important for one than the other, and the way it impacts the inverse square law (and especially the fact that it would work better for one than another, as one is further away from steady state) is not trivial to me.

We have removed 'identical' to avoid any ambiguity, such that the sentence now reads "...can observe contrasting tipping behaviours for systems that differ only by the strength of the linear restoring force proportionality factor, ..."

c11: L224(and after) - the dynamical variable in the Cessi model is not the change in salinity flux, but

the meridional salinity gradient. It is very confusing as the forcing applied to the model is actually the meridional gradient of freshwater flux (for example in L259).

Thank you we have now corrected this so that we refer to the dynamical variable as the "meridional salinity gradient" and at the first use of "freshwater flux" we write "...meridional gradient of freshwater flux, hereon in referred to simply as freshwater flux, ...".

c12: L305 - I do not see the asymmetry of the orange band in Fig 8.a, I must have missed something?

The asymmetry can be seen by the orange line not being at the centre of the orange banding. We have now made the following revision to make this clearer "Note that the nonlinear relation between the diffusive timescale parameter and the threshold location makes the orange band not symmetrically distributed around the orange line."

s1: L5 - I would replace "and avoid tipping" by "without tipping" (it can sound like the fact of overshooting in itself makes a tipping less probable).

We have updated the text as suggested.

s2: L7 - I would add "However, in the real world ...".

We have updated the text as suggested.

s3: L10 - I would write "separating safe from unsafe overshoot", for the same reason as s1.

We try to avoid using "safe" and "unsafe" because there are risks associated with any overshoot that we do not consider. However, we also acknowledge the issue with "avoid" and so we now write "... boundary separating overshoots without tipping from those that do tip.". We have made similar modifications throughout the manuscript.

s4: L16 - "in this conceptual model, ..." would be more straightforward than "In our study, ...".

We have updated the text as suggested.

s5: L123 - I would find it more natural to have "We now illustrate ..." at the beginning of the next paragraph.

We have moved the sentence to the start of the next paragraph as suggested.

s6: L179 - I would remove "quite".

We have removed the word "quite".

We are grateful for the constructive reviewer comments received on our manuscript. These comments are repeated in black, and our responses given in blue.

**Response to Reviewer 2**

**Summary**

The study by Lux-Gottschalk & Ritchie explores how parametric uncertainty introduces and impacts uncertainty in the response of tipping elements to overshoot trajectories, which temporarily exceed the critical tipping threshold.

First, the impact of uncertainties in the locating of the critical threshold and the linear restoring rate on the response of tipping elements to overshoot trajectories of varying overshoot magnitudes and durations is illustrated based on a simple dynamical system with a fold bifurcation (Sect. 2.1 and Sect. 2.2). In particular, the tipping element response is described in terms of probabilities of tipping, based on a modification of a previously derived inverse square law (Ritchie et al. 2019). Therein, the uncertainty in the location of the critical threshold leads to a larger uncertainty in the tipping response to overshoot trajectories than the linear restoring rate. Constraining the uncertain parameter can reduce uncertainties in this tipping response. In addition, the effect of changes in the mean vs the range of the distribution of the uncertain parameter is explored. In a next step (Sect. 2.3), this framework is extended to a box model of the Atlantic Meridional Overturning Circulation with a similar bifurcation structure and an uncertain parameter (diffusive timescale) that concurrently translates into an uncertain location of the critical freshwater flux threshold and the linear restoring rate.

**General comments**

The paper approaches the relevant and scientific interesting question of uncertainty in the tipping response to overshoot trajectories in the presence of uncertain parameters from a theoretical / conceptual side. The results could perhaps serve as a starting point for further (probabilistic) assessments of potential critical transitions in the Earth system.

The introduction contains relevant information on tipping and the concept of 'safe' overshoots. In some places, it seems repetitive or related information is spread across the introduction. Some restructuring of the introduction might help to avoid these repetitions and it could also support making the motivation for and main aims of this study clearer. For example, tipping is introduced and defined in L19-25, and some additional general information is added in L52-54. Impacts of tipping are mentioned in L26, and in L45-46 the potential consequences of AMOC tipping are described. Uncertainties are first referred to in L32-40, and laid out in more detail in L77-82.

We have restructured the introduction. Thereby, we improved on grouping together the introduction of tipping phenomena (including moving the discussion on the interplay of the tipping mechanisms to the Conclusions section), the impacts of tipping, and the introduction of uncertainties. We also removed redundancies to minimise any repetition.

While the approach of explaining the basic interplay between the overshoot trajectory and the location of the critical threshold, the linear restoring rate or the diffusive timescale (in the AMOC box model) in the beginning of each section before introducing uncertainty in these parameters is a good starting point for understanding the key figures displaying tipping probabilities, it is not always easy to follow the results presented in this study. This is for two main reasons:

1. The clarity of the language could be improved throughout the manuscript. In some places, the language distracts from the key messages of the paper. These key messages might be formulated more concisely. Please also see the specific comments below.

2. The methods are not given in the main manuscript but described in the Appendix. To allow the reader to properly follow the results, the models and the derivation of the tipping probability based on the (modified) inverse square law and numerical simulations should be described before presenting

the impact of uncertainty in the location of the critical threshold and the linear restoring rate on the tipping response. In addition, the Bayesian Inference to constrain uncertain model parameters should be explained in such a section.

We have given careful consideration to how much of the appendix can be moved to the main manuscript without disrupting the flow of the text too much. Specifically, we now introduce the respective forcing profiles and the models at their relevant locations. We have also moved the Bayesian Inference section to the main manuscript as suggested as we agreed that it's an important approach that we introduce as a method for constraining parameter uncertainty. More explicit details about the probability of tipping have also been provided in the main manuscript where we write

"For any given overshoot profile, there will be a critical threshold location that separates tipping (lower thresholds) from not tipping (higher thresholds). Therefore, the cumulative probability density function at this critical threshold gives the probability of tipping. This probability of tipping, or more precisely the critical tipping threshold location, can either be calculated numerically (using a bisection method) or via a modification to the inverse square law."

Here, we have additionally moved the modified theory that accounts for time over an arbitrary threshold to the main text, but have decided against moving the full derivation as not to disrupt the flow too much and therefore instead refer the reader to appendix A1 for the full derivation.

The manuscript illustrates how parametric uncertainties translate into uncertainties in the tipping response based on conceptual models, and I acknowledge that a lot can be learnt from these approaches, e.g. to showcase that constraining parametric uncertainty is important to constrain the tipping response to overshoot trajectories. It could be interesting to expand the discussion of if / to what extent / how this framework and the conceptual results could be applied to more complex situations.

In the paragraph on Bayesian inference in the discussion section, we expand on the potential of our methodology to be carried over to a more complex AMOC box model. The benefit is that for this box model, box-averaged time series data of a general circulation model can be used for the Bayesian inference instead of just synthetic data. We also emphasise on the challenge of dealing with the additional Hopf bifurcation (and potentially rate-induced tipping) occurring for some model parameter configurations.

I have included more specific comments, questions and suggestions below.

**Specific comments**

L01: Maybe replace 'Earth' by 'Earth system' or 'climate system'. In addition, can these subsystems be considered to currently be in a stable (steady) state?

Good point, we have reformulated the first sentence of the abstract to read "Many subsystems of the Earth system, that are currently in a stable state, are at risk of undergoing abrupt transitions to a drastically different, and often less desired, state due to anthropogenic climate change."

L03: Maybe replace 'common mechanism of tipping' by 'possible mechanism of tipping'.

We have updated the text as suggested.

L05: Maybe replace 'is possible' by 'could be possible'.

We have updated the text as suggested.

L07: Please give references in brackets here and elsewhere if not integrated into the sentence, i.e. (Ritchie et al., 2019).

References now appear correctly formatted throughout.

L07: What is meant by 'other system features'? Please specify.

We have now added "such as inherent timescales" to indicate that this is one additional system feature that plays a key role in determining the response to an overshoot.

L08: I would like to suggest to add 'in response to a temporary overshoot' (or similar) after 'probability of tipping'.

We have updated the text as suggested.

L14-15: This statement appears as a key result / conclusion of the analysis in the abstract. I was wondering whether this is the case, e.g. when reading L166-168, and also what it implies for conclusions on mitigation pathways (as indicated in the previous sentence).

This is indeed a key conclusion of our analysis, that the tipping behaviour for the same overshoot pathway might completely change based on the understanding of system characteristics. For instance, this is the purpose of panels (a) and (b) in Figures 1, 3, and 6. We have modified this sentence to make this point clearer "A pathway believed to offer little danger of tipping, may become highly dangerous if the tipping threshold were to be closer than previously understood."

L16-17: To me, the sentence suggests that many parameters in the box model have been tested and the diffusive timescale has been identified as particularly relevant for determining uncertainty in the AMOC tipping behaviour. I would like to suggest to rephrase to: 'In addition, we illustrate how constraining the highly uncertain diffusive timescale within this box model reduces the tipping uncertainty of the AMOC in response to overshoot scenarios' (or similar).

We have largely updated the text as suggested, which now reads "In this conceptual model, we illustrate how constraining the highly uncertain diffusive timescale (representative of wind-driven fluxes) within this box model reduces the tipping uncertainty of the AMOC in response to overshoot scenarios."

L20: 'sudden' could suggest that tipping occurs abrupt in time, which may not necessarily be the case. Please consider to reformulate.

We have replaced "sudden" with "abrupt" and rephrased as follows "Tipping events are abrupt transitions that may occur if some external forcing crosses a critical threshold (Scheffer et al., 2012)."

L22: 'kicked off their current stable state' could refer noise- and rate-induced tipping, but not necessarily bifurcation-induced tipping, which is associated with the loss of stability beyond a critical threshold in some control parameter, followed by a critical transition to a remaining stable state. Please consider to reformulate.

Agreed, we have reformulated the sentence to now read "Systems that are currently in a stable state may find their state ceasing to exist and therefore causes the system to transition (potentially irreversibly) to a drastically different state (Lenton et al., 2008).".

L28: Maybe replace 'are assumed' by 'have been suggested' (or similar).

We have updated the text as suggested.

L29-31: Maybe replace 'can still be avoided' by 'could potentially be avoided'. I would also like to suggest to explicitly acknowledge that this statement is based on model simulations.

We have reformulated the sentence to read "It is important to note that for some elements, climate model simulations have shown that tipping might still not occur, if the reversal of the forcing is sufficiently fast (Jackson et al., 2022)"

L55-72: I am not sure how helpful the description of the different mathematical mechanisms of tipping is, in particular given that the relation to overshoots is not directly described. Please consider to connect the different mechanisms to overshoot trajectories. This may also help to justify the focus on bifurcation-induced tipping in this paper as stated in L73.

We shortened our elaborations on other tipping mechanisms in the introduction. Moreover, we now motivate explicitly our focus on bifurcation-induced tipping: "Since we would like to understand overshoots of critical thresholds of global warming, here, as a natural restriction, our primary focus is on better constraining the mitigation window for bifurcation-induced tipping".

L88-89: Please check the section numbering in the Appendix. In addition, I would also like to suggest to move (parts of) the Appendix into a section that focuses on the Methods in the main part of the manuscript (see general comments). This might be helpful for readers that are not familiar with the fold bifurcation and the AMOC box model, and would allow to remove some of rather detailed information of the AMOC box model in the introduction (L86-89). This also applies to the probabilistic extension of Ritchie et al. (2019) and the inverse square law, as its development is described as a main aim of this paper (L90).

We have decided to move parts from the Appendix to the main body of the manuscript. We have done a substantial restructuring to integrate all of the Appendix in the main body except for the details of the derivation of the overshoot theory for an arbitrary threshold and the parameter specifications for the Stommel-Cessi model. Moreover, we now put the rather detailed information on the AMOC box model at the end of the introduction together with the actual equation for the dynamics exhibiting the fold bifurcation in a separate section "Problem Setup".

L91: Maybe replace 'fold tipping scenario' by 'tipping via a fold bifurcation' (or similar).

We have updated the text as suggested.

L119: Why is the location of the critical threshold the 'biggest uncertainty to consider'? Is this already a result of the analysis, based on the argument that follows this statement, or motivated by previous studies?

We have removed this claim here, and instead write "First, we consider uncertainty in the location of the tipping threshold.". Although the location does determine if an overshoot of the threshold occurs or not and therefore will have particular influence for slow overshoots. This is already mentioned when discussing the results and on reflection do not need to make such a statement without first showing the results. Therefore the remainder of the old paragraph has been deleted and replaced with more experimental detail (see later comment).

L126: Maybe replace 'large' by 'far'.

We have updated the text as suggested.

L130: 'early' could be associated with a time dimension. Please consider changing 'early' to 'low' (or similar).

We have updated the text as suggested.

L131: I am not sure about the formulation 'otherwise identical system'. The bifurcation diagrams also differ in e.g. the location of the basin boundary in addition to the location of the critical threshold.

We have now removed "but otherwise identical". The location/distance to the basin boundary does indeed

differ for the same forcing level. However, critically the distance is the same for the same distance from the threshold. We have now added further details to this effect in the paragraph above by writing "We will use the prototpyical fold model (2) and only consider different parameter values for the tipping threshold location, $p_b$. Note, that the linear restoring force (and distance to the basin boundary) will be different at the same forcing level, for two different threshold locations. Importantly however, all system characteristics remain unchanged when the system's are the same distance to their respective thresholds."

L133: Maybe replace 'a threshold further away' by 'a higher threshold' (or similar).

We have updated the text as suggested.

L136: I would like to suggest to use a different terminology for cases without tipping. In the manuscript, formulations such as 'tipping can be avoided' are often used when there is no critical transition due to a high tipping threshold (or similar; see e.g. L158-159, L177, L185-186, L307-308, L323). This formulation could be misunderstood.

We have carefully gone through the manuscript and rephrased instances of 'tipping can be avoided' and similar. Although we would like to highlight that the the L307-308 instance the reviewer mentions does not refer to 'tipping' and instead mentions 'avoid the system crossing the threshold', and so this has been left unchanged.

L137-138: I am not sure whether it is clear how a tipping probability is assigned here. Please see the related comments on including a section on the methods in the main part of the paper.

We agree and have now taken the relevant text from the appendix and added it to the main manuscript (and expanded on it), just before introducing Figure 1(c). Please see previous comment for more details, but essentially, for any overshoot there will be a critical parameter value (e.g. threshold location) that separates tipping from not tipping. Where this critical value falls in the probability distribution for the parameter determines the probability of tipping (i.e. from the cumulative probability density).

L139: While it may be clear to some readers, it might still be helpful to add a short explanation for the choice of this range for the location of the critical threshold.

The range is arbitrarily chosen. However, we now explain that different tipping behaviour has been observed for a low and high threshold and so will now proceed with looking at all possible values between these levels. We write "If we now extend this to a continuous range between these tipping threshold locations (initially all locations are assumed to be equally likely), then a tipping probability can be determined based on this arbitrarily chosen uniform distribution, $p_b \sim \mathcal{U}[2.0, 2.3]$."

L140: I would like to suggest to indicate the quantities on the axis of Figure 1(c) (that is, the time spent over the lowest possible threshold and the peak in external forcing) in Figure 1(a). In addition, maybe replace 'based on' by 'depending on'.

Thank you for the suggestion We have now added a black arrow and dashed lines to indicate the peak external forcing and a green shaded region for the time over the lowest threshold in Figure 1(a). This has been applied consistently throughout all similar figures.

L142-149: Please indicate the probability ranges (in terms of the shading and the black lines) in Figure 1(c) as an additional legend, as in Figure 8(b). Figure 1(c): Is there a reason why the colouring stops at tipping probabilities of 50% and larger, in contrast to Figure 8(b) and given that for a known threshold (without uncertainties) there would only be one line separating tipping (100% probability of tipping) from non-tipping (see also e.g. L310-312), if I understand correctly.

We have added a legend to this plot and all others to indicate the probability ranges. We consider so called

‘negative’ tipping points that we would like to avoid. Therefore, it is more important to focus on the lower probability levels than high probabilities. However, we do now add a 99% probability level for Figure 4(b) to help explain the reduction in tipping uncertainty that is not so clear when just considering the lower probability levels.

L150-155: It could be helpful to describe more explicitly in the beginning of the description of the results how uncertainty in the response to overshoot trajectories and a low / high tipping probability relate. Only at the end of Sect. 2 (L334-335) a higher uncertainty is linked to a tipping probability moving closer to 50%.

We have now added the following paragraph to address the link between tipping probability and uncertainty in tipping behaviour: "The distance (along cross sections of either the peak forcing or time over threshold) between the tipping probability contours provides an indication of the uncertainty in the tipping behaviour. A large distance between the probability contours reflects a high uncertainty in tipping behaviour. Therefore, the performance of reducing uncertainty in the tipping behaviour will be determined by the ability to reduce the distance between the probability contours upon constraining uncertainty in the system characteristics."

Note, there will always be overshoot trajectories that result in 50% tipping probability, but provided the uncertainties in the model are constrained these signify the boundary that separate overshoots that cause tipping versus those that do not.

L151: Please consider rephrasing to ‘peak forcing not even overshooting the threshold’ (or similar) instead of ‘profile not even overshooting the threshold’.

This has been rephrased to "peak forcing not even exceeding the threshold".

L152-153: I am not sure about the structure of this sentence. Please check.

We have rephrased the sentence to read "Though for lower thresholds, an overshoot of the threshold will occur and therefore the reversal in the forcing needs to be sufficiently quick to ensure tipping does not happen."

L161: Please specify ‘distribution’ here and elsewhere (e.g. L209). I assume that in this case it is referred to the ‘distribution of the location of the critical threshold’.

We have updated these instances which refer to the threshold location and restoring force proportionality factor respectively.

L166-168: I was wondering whether it might be possible to align the (description of the) probabilities with the language used in the IPCC.

We have now aligned the probabilities with the language used in the IPCC where possible, by writing "The black dotted contour corresponds to a 1% probability of tipping (*exceptionally unlikely* in IPCC terminology (Masson-Delmotte et al., 2021)); dot-dash a 10% probability of tipping (*very unlikely*); dotted 25% probability of tipping; and solid 50% probability of tipping (tipping becomes *more likely than not*)."

L171: ‘however’ could be removed. Maybe replace ‘factor’ by ‘parameter’ or ‘system characteristic’ (or similar).

We have removed the word "However" and replaced "factor" by "system characteristic".

L171-172: I would suggest to decide for one expression here. In addition, it might be helpful for the reader to give a short explanation of the strength of the linear restoring rate. Is this different or the same for both stable equilibria or is this a general system property (as indicated by the e.g. expressions ‘weaker system’, ‘strong system’; L182-187)? This could also be included in a dedicated section describing the methods in the main part of the manuscript.

We now consistently use "restoring force" throughout the manuscript. The restoring force will affect both stable equilibria but we are primarily referring to the initial state the system resides in. We have now re-phrased the sentence to read "Another important system characteristic for determining the mitigation window for overshoots is the strength of the linear restoring force – decay rate towards the stable equilibrium after making a perturbation to the system." Also, instead of 'weaker system' we now refer to the 'system with weak restoring force'.

L176-177: I am not sure about the formulation 'identical systems', as the systems also differ in e.g. the location of the basin boundary. How does the basin boundary relate to the restoring rate?

We have removed 'identical' to avoid any ambiguity, such that the sentence now reads "...can observe contrasting tipping behaviours for systems that differ only by the strength of the linear restoring force proportionality factor, ...". The basin boundary also moves further away from the initial stable state for weaker restoring force proportionality values, which aids recovery. Please see your comment and our response regarding 'curvature of the fold' for more details.

L179: Maybe replace 'boundary for the basin of attraction' by 'boundary of the basin of attraction'.

We have updated the text as suggested.

L181: Which stable state do you refer to here (i.e. the upper stable state or the lower stable state or both)?

We are referring to the upper branch, though note this is the only stable branch. However, to remove any ambiguity we now write "The system with the weaker linear restoring force has a weaker 'pull' towards the stable branch the system starts at, ..."

L183-184: What is meant by 'curvature of the fold'? I am not sure whether this becomes clear without a brief explanation.

The curvature of the fold refers to the curvature of the equilibrium curves (unstable and stable branches together) at the fold point. However, the importance of this is that it changes the distance to the basin boundary and so we now refer to this instead of the 'curvature of the fold' by writing "An additional advantageous effect of a weaker restoring force proportionality factor is that the the boundary of the basin of attraction is further away (from the initial stable state). Therefore, the system with weak restoring force (thin and opaque black dashed curve) can cross the basin boundary at lower system state values than compared to the system with a strong restoring force (thick and translucent black dashed curve)."

L184: What does 'earlier' refer to here? How is time involved?

Please see comment above. We have now rephrased sentence such that we remove any suggestion of 'time'.

L198: Please consider rephrasing to: 'the tipping uncertainty is substantially smaller' (or similar). Otherwise the sentence could suggest that an uncertain restoring force is effective in 'reducing uncertainties'.

Good point, sentence modified as suggested.

L205: I think the comma after 'tipping' is not needed here.

Thank you for spotting this. The comma has been removed.

L206: Please consider rephrasing to: 'An overshoot trajectory that sits...' (or similar).

We have taken the suggestion as given and also removed the ambiguity regarding 'initial' and 'now' as noted in other comments, such that the sentence now reads "An overshoot trajectory that sits on the 50% probability of tipping curve based on the initial distribution, would be considered to be very unlikely ($< 1\%$) to cause tipping given the knowledge-based distribution."

L208-209: This sentence is unclear to me. What is meant by 'the separation between the 1% and 50% curves'?

The separation refers to the distance between the 1% and 50% curves both for cross sections along the x and y axes. We have rephrased the sentence to make this clearer: "However, unlike for the threshold location, the distance (both horizontally and vertically) between the 1% and 50% tipping probability curves for the initial and knowledge-based distributions of the restoring force proportionality factor have barely changed.".

L210: Please specify 'uncertainty' here and elsewhere. It should be clear whether it is referred to uncertainty in tipping or uncertainty in e.g. the location of the critical threshold.

We now state that this is a "decrease in parameter uncertainty." and checked the remainder of the manuscript to ensure that all 'uncertainty' is now properly attributed.

L210: Maybe I have misunderstood this sentence, but to me, it looks as if the alternative distribution covers a different range of the restoring force proportionality factor (3.5-6.5) than its initial distribution in Figure 4(a).

We acknowledge the phrasing needed improving here. We meant that the width of the distribution was the same, but indeed covers higher values. We now write "To illustrate this, we consider an alternative uniform distribution, $\kappa \sim \mathcal{U}[3.5, 6.5]$ (blue), that is of the same width but has a much higher mean for the restoring force proportionality factor, see Figure 4(a)."

L211: What is meant by the 'width of the banding'? Do I understand correctly that this refers to the 'distance' between the 1% and the 50% probability curves? In general, it might be helpful for the reader to introduce and use a clear terminology when describing Figure 4(b) (and similar figures). So far, different notions have been used throughout the manuscript (e.g. uncertainty in tipping behavior, probabilistic critical boundaries, uncertainty in the tipping probability, uncertainty in the probability of an overshoot avoiding tipping).

Yes exactly, the 'width' is indeed the 'distance' between the 1% and 50% probability contours. The text has been changed accordingly. Further, as discussed in a previous comment, we have added a paragraph to define the links between the tipping probabilities and uncertainty in tipping behaviour. Consistency across the manuscript has been improved.

L212-213: This sentence is not complete. Please rephrase.

We have rephrased the sentence to read "Therefore, this illustrates that an uncertainty in the restoring force for large values is less critical than at lower values."

L215-217: These sentences are not clear to me. If I understood correctly, a clear boundary that separates tipping from non-tipping exists for a known restoring rate without uncertainties, and would be given by the inverse square law. What does '(uncertainty of) the location of critical boundary (separating tipping from not tipping)' refer to? How can there be 'little change in the width' and a decrease in the 'uncertainty of the location of the critical boundary' at the same time? Please also see the related comment for L211.

We have clarified in the text that it is "... little change in the width between the 1% and 50% boundaries is observed." We now also include in Figure 4 the 99% boundaries to show that the uncertainty in the location of the critical boundary decreases by writing "This can be seen by plotting the 99% probability of tipping boundary, and noticing that the separation between the 1% and 99% boundaries does indeed reduce."

L231: Please consider rephrasing to: 'The chosen ranges of the advective and diffusive timescales correspond to...' (or similar).

We have updated the text as suggested.

L235-259: This is an extensive description of critical freshwater fluxes depending on the advective and diffusive timescales. I would suggest to shorten these paragraphs, if possible, and focus on the aspects that are directly relevant for the overshoot scenarios.

We agree that there is a detailed description of the critical freshwater flux thresholds. However, we think it is important to highlight the differences between non-dimensional and dimensional quantities. Specifically, for non-dimensional quantities it has been shown that the freshwater flux threshold moves to lower values for decreasing the ratio of timescales. However, due to the timescale scaling on the non-dimensional freshwater flux, the dimensional freshwater flux threshold can move to lower or higher values depending on whether the change in the ratio is caused by changing the advective or the diffusive timescale.

L238: I would like to suggest to add the missing bistability as the main reason (if I understood correctly), which then means that no critical freshwater flux exists.

We have updated the text accordingly by now writing "no tipping is possible as there exists no bistability region and hence no critical freshwater flux and so the corresponding region is coloured white".

L257-259: These sentences are unclear to me. What is meant by 'the upper critical threshold moves later'? Does the second sentence refer to the lower critical freshwater flux threshold?

We were meaning that the critical thresholds move to higher freshwater values and yes, the second sentence does refer to the lower critical freshwater flux. Although to remove ambiguity we refer to the upper and lower critical freshwater flux thresholds as the critical freshwater flux for AMOC collapse and recovery respectively. Here we now write "Concurrently, both critical thresholds move to higher values, so these factors alone will make tipping less likely for any given overshoot. Additionally, the freshwater flux can be stabilised at a higher level, such that only the AMOC on state exists (since the threshold for AMOC recovery moves higher)."

L261: Please add a short justification for a stabilization of the freshwater flux at 0.25 Sv.

The stabilisation level is chosen such that in as many cases as possible the freshwater stabilises in the bistability region. We have now added "The stabilisation level ($p_s tab$) is chosen such that the freshwater always stabilises below the critical threshold for AMOC collapse, but above the threshold for AMOC recovery in most cases.".

L263: I am not sure whether a small diffusive timescale should be referred to as an 'event'. I would like to suggest to rephrase to 'In the case of a small diffusive timescale, the tipping threshold...' (or similar).

We have updated the text as suggested.

L263: 'late' (and 'early') could be associated with time. Please check the formulation here and elsewhere.

Here, we now use "far away" and elsewhere or "higher" (and "lower").

L266: Why is the exact number of 0.2 Sv given here? If I understand Figure 6(b) correctly, the AMOC would recover for a reduction in the freshwater flux to below approx. 0.22 Sv. Please also add for which value of the diffusive timescale this threshold is given.

We were providing an example level but indeed any level below 0.22 Sv would enable AMOC recovery.

To remove this ambiguity we now write "Note also that the bistability region is small for small diffusive timescales and therefore, in this example ($t_d = 455$), the AMOC would recover (regardless of the overshoot time) if the freshwater flux is reduced back to below the lower fold at approximately $0.22\ Sv$."

L272: Maybe it could be referred to the figure that shows the prior distribution of the diffusive timescale.

We now write within the bracket "(based on the prior distribution for the diffusive timescale given in Figure 7(a))".

L273: Maybe replace 'recovery of the AMOC on state' by 'recovery to the AMOC on state'.

Agreed, we have changed the wording to "recovery to the AMOC on state".

L278-283: What is meant by 'initialising the simulations in equilibrium' in this context? I think some more explanation may be helpful for the reader here to understand how this acts as an additional 'source of error'. Does 'error' refer to the discrepancy between numerics and theory? Please specify. In addition, how is the applicability of the framework presented here to more complex models effected by the discrepancy between numerics and theory?

Strongly forced systems are not in equilibrium. Therefore, by assuming the AMOC is in equilibrium at the start, will cause an overestimation of the numerical tipping probability and will be particularly relevant for thresholds that are close. This has been clarified in the text by writing "Specifically, strongly forced systems are not in equilibrium. Therefore, making the assumption that the AMOC is in equilibrium at the start of the simulation, will cause an overestimation of the numerical probability of tipping particularly for tipping thresholds that are close." The applicability of the framework for more complex models has received more discussion in the Conclusions section, please see later comments. However, it does not fit here as we are discussing the discrepancies between the numerics and theory for our example.

L288: Maybe change to 'substantial' or 'strong' instead of 'dramatic'.

We have made the change to "strong".

L289-290: Does 'previously' refer to the prior distribution? This sentence is not clear to me. This also applies to 'previously' (and 'now') in L294.

Yes, indeed it does, and we acknowledge the ambiguity and so we now rephrased the two sentences to read: "An overshoot that has a 25% probability of tipping, based on the prior distribution, would be classified as *very unlikely* with less than 1% probability of tipping given the posterior distribution". and "This ultimately means that, if the tipping probability is 50%, given the prior distribution knowledge, this will change to a probability of tipping that is greater than 99% for the constrained posterior distribution (not shown)."

L290: How is the 'stabilisation level' defined? I assume that it refers to the forcing level after the overshoot, but a clear definition of this term is missing (or maybe I have missed it).

Indeed, the stabilisation level corresponds to level of forcing after the overshoot. We now include the equation for the forcing in the main manuscript (equation (10)). There we state that the forcing finishes at $p_{stab}$.

L293: I am not sure if 'guaranteed' fits well here.

We have reformulated and now write "... instead of observing a recovery back to the AMOC on state."

L294-295: It might be helpful to show the >99% probability of tipping (see previous related comment on the probability levels that are shown in the figures).

Please note that the 99% contour does not exist for the prior distribution since the stabilisation level for the smaller diffusive timescales is below the threshold for AMOC recovery (i.e. stabilises in regime where only the AMOC on state exists). However, we have re-phrased the sentences to not refer to the 99% contour and avoid any confusion. This now reads "Note that, the critical freshwater flux threshold, for AMOC recovery, is only below the stabilisation level for diffusive timescales greater than 400 years.Thus, for the posterior distribution, the time taken to reverse the freshwater flux is critical to determine whether tipping occurs or not. Whereas previously, given the prior distribution knowledge, the probability for AMOC recovery would be non zero regardless of the time taken to reverse the freshwater flux."

L332: The comma after 'distribution' may not be needed.

Indeed the comma is not needed and so has now been removed.

L332-335: If possible, it might be helpful to indicate this example in Figure 8(b).

We have now added arrows to indicate the scenarios of reducing parameter uncertainty while keeping the mean fixed (green) or the mean increasing (red).

L335-336: Can this also be inferred from a specific figure?

Not easily. However, we have further explained that we are establishing that the this particular overshoot is close to the critical overshoot that separates tipping from not tipping. We now write "However, here the orange line is moving down towards the black line and at the same time the banding is shrinking around the orange line. Therefore we are instead establishing that this particular overshoot is close to the critical overshoot that separates tipping from not tipping, but importantly, if all possible overshoot trajectories are considered then the overall uncertainty in the probability of tipping will still reduce.".

L338: Maybe replace uncertainty on model parameters' by 'uncertainty of/in model parameters'.

Changed to "uncertainty in model parameters"

L340-354: In the summary of the role of the location of the critical threshold and the linear restoring rate for the system response to overshoot trajectories, the impact of uncertainties in these parameters on the tipping response could be addressed in more detail (as this is the focus of the paper).

We have now added the following sentence to the opening paragraph of the Conclusions section "Specifically, we have found that the tipping behaviour from a single overshoot scenario can completely change based solely on either the location of the tipping threshold or the strength of the linear restoring force.". Otherwise, we have three paragraphs summarising the role of these system characteristics, but have edited them to make them more focused on the impact of uncertainties.

L343: Maybe reformulate and replace 'further away' by 'high' (or similar).

Sentence has been modified such that it now reads "Assuming an overshoot of the threshold does occur, both the peak overshoot distance and the time spent over the threshold would be smaller for a high threshold compared to a low threshold."

L346: Something may be missing after 'especially the time over'. Please check.

We have rephrased the sentence to read "Minimising the peak distance and especially the time duration of an overshoot, reduces the likelihood...".

L352: Maybe I misunderstood something, but I got confused by 'quasi-static'. Do the black lines in e.g. Figure 6(b) correspond to equilibrium states or quasi-equilibrium states?

The black curves correspond to the equilibrium states of the static (or frozen) system. We have re-phrased the sentence to remove any ambiguity "This can be intuitively understood by a weaker restoring force causing the system to further lag the stable equilibrium (of the static system) under a change in external forcing."

L360-362: I would suggest to combine these two sentences into one sentence, e.g. 'Constraining parameter uncertainty, for instance by performing Bayesian inference on observational data can greatly reduce...' (or similar).

We have combined the two sentences as suggested.

L362-363: Here, it could be helpful to discuss the applicability of the (probabilistic) inverse square law relationship in more complex models (i.e. beyond the conceptual models used here) in more detail.

We have now discussed the potential of our techniques to be carried over to a more complex five box AMOC model with the opportunity of using general circulation model data instead of synthetic data. We pointed out the remaining challenge of dealing with an additional Hopf bifurcation (and potentially rate-induced tipping), for which the inverse square law theory as it is does not hold any more. Please see also our response to your general comment just before the specific comments start.

L366-367: While the concept is interesting, I am not sure whether the introduction of an 'overshoot budget' is needed here. It is not used beyond this paragraph, and even in this paragraph it is not entirely clear to me how it relates to the main part of the manuscript.

On reflection, we agree that the 'overshoot budget' is not critical to the main part of the manuscript and so we have removed the paragraph.

L374-375: This sentence is not complete. Please also consider putting this statement (and a similar statement in L367-369) into context – it might still be favourable to avoid overshooting critical thresholds at all, if possible, instead of 'favouring large and short overshoots as opposed to small and long overshoots'.

The sentence has been rephrased to indicate that when considering variability as well it is especially critical to minimise the duration of an overshoot. We now write "This further emphasises that minimising the duration of any overshoot of a tipping threshold is paramount to preventing tipping." The other sentence referred to has been deleted along with the paragraph about an overshoot budget, please see comment above.

L375-377: Is the need to constrain system uncertainties in the case of rate-induced tipping specifically related to overshoot scenarios or does it apply in general?

Here, we were particularly referring to overshoot scenarios for systems also capable of undergoing rate-induced tipping. To remove the ambiguity we have now added the following to the end of the sentence "... and therefore constraining system uncertainties becomes even more critical when considering overshoot scenarios."

L381: I am not sure whether a 'planned overshoot' fits well as a formulation. Please consider rephrasing.

We have rephrased the sentence to now read "For example, the precise peak overshoot distance and time spent over the tipping threshold is likely to be uncertain."

L383: 'AMOC tipping is very unlikely' is a strong statement, based on the AMOC box model. While limitations are discussed in the following sentences, I would still like to suggest to reformulate and clearly indicate that this statement refers to the box model.

We have modified the sentence such that it now reads "... and therefore this low dimensional box model

would suggest AMOC tipping is very unlikely."

L389: Maybe add 'in response to possible overshoot trajectories' at the end of the sentence.

We have updated the text as suggested.

L389-390: 'elements tipping'?

We have now amended to read "... elements of the climate system tipping ...".

We are grateful for the constructive reviewer comments received on our manuscript. These comments are repeated in black, and our responses given in blue.

**Response to Reviewer 3**

The manuscript "Uncertainty quantification for overshoots of tipping thresholds" by Lux-Gottschalk & Ritchie looks at how uncertainties affect the probability of tipping. Motivated by the uncertainty of real-world models, they first conduct probabilistic assessments of how uncertainty within parameters can lead to uncertainty in the tipping response to overshoot trajectories. Their general framework is then highlighted in a box model of the Atlantic Meridional Overturning Circulation which has a similar underlying structure. Overall, this is good and interesting work, and it fits well into the scope of ESD. Additionally, this work has the potential to be the basis for more complex scenarios. Below I list some comments that should be considered before publication.

**Specific comments:**

As I was reading the manuscript, initially I had a hard time following what was happening as I was flipping back and forth to the appendix from the main text. While I understand why the authors put together the appendix, I think it would help the strength of the paper to present or at least describe in more detail many of the methods and derivation of the tipping probability from the inverse square law, the models, etc. It may be worth considering just putting some of the appendix within the main text rather than it being separate as it will help with readability. Another alternative is to better reference the appendix when needed.

We have given careful consideration to how much of the appendix can be moved to the main manuscript without disrupting the flow of the text too much. Specifically, we now introduce the respective forcing profiles and the models at their relevant locations. We have also moved the Bayesian Inference section to the main manuscript as it's an important approach that we introduce as a method for constraining parameter uncertainty. More explicit details about the probability of tipping have also been provided in the main manuscript where we write

"For any given overshoot profile, there will be a critical threshold location that separates tipping (lower thresholds) from not tipping (higher thresholds). Therefore, the cumulative probability density function at this critical threshold gives the probability of tipping. This probability of tipping, or more precisely the critical tipping threshold location, can either be calculated numerically (using a bisection method) or via a modification to the inverse square law."

Here, we have additionally moved the modified theory that accounts for time over an arbitrary threshold to the main text, but have decided against moving the full derivation as not to disrupt the flow too much and therefore instead refer the reader to appendix A1 for the full derivation.

I think there needs to be a clearer description on how uncertainty/overshoot trajectories/tipping probabilities all relate, and the motivation needs to be more clearly depicted in the introduction.

We have carefully restructured the introduction to better group togethter the introduction of tipping phenomena, the impacts of tipping, and the introduction of uncertainties. Moreover, we added an explicit motivation to restrict ourselves to bifurcation-induced tipping in our manuscript: "Since we would like to understand overshoots of critical thresholds of global warming, here, as a natural restriction, our primary focus is on better constraining the mitigation window for bifurcation-induced tipping".

L19: There is a misspaced apostrophe. Also, this is the start of your introduction, and it reads awkwardly/a bit unclear. I suggest combining the first two sentences and making it more clear how tipping points relate to the sustainable development goals/climate action.

We have now removed the first sentence.

L20-21: Tipping events are not only in environmental areas or climate subsystems. While I recognize it is environmental for the context of this work, it is misleading to say tipping only occurs for environmental conditions.

We have modified the sentence so that instead of 'environmental conditions' we now use 'external forcing'.

L28: Replace the word assumed with proposed/suggested, etc.

We have replaced the word "assumed" with "suggested".

L55-73: While I appreciate the details on the three main tipping mechanisms and a nod/summary to the work that has been conducted, I am not sure if an entire paragraph is needed. How does it relate to your work? You mention that you consider various profiles that have different impacts on the overshoot of the critical bifurcation threshold value. If there is a clearer connection regarding each tipping mechanism to overshooting, then make those connections. Otherwise, you do not need all of this.

We agree that the level of detail for the other tipping mechanisms was too elaborate given the focus of our manuscript. Therefore, we have significantly shortened our elaborations in the introduction, including moving the interplay between mechanisms to the conclusions section where it is specifically discussed. Moreover, we now motivate explicitly our focus on bifurcation-induced tipping.

L109: Bayesian statistics is referenced incorrectly. There doesn't seem to be a Materials and Methods section? Then in the appendices themselves, you have two A1's.

Thank you for spotting this. However, as described in a previous response we have decided to move the Bayesian Inference section to the main manuscript.

L119: Arguably, the biggest uncertainty to consider is the location of the tipping threshold. Why? You seem to just drop this at the start of 2.1. It would be helpful to say where this is coming from or foreshadow it.

We have removed this claim here, and instead write "First, we consider uncertainty in the location of the tipping threshold.". Although the location does determine if an overshoot of the threshold occurs or not and therefore will have particular influence for slow overshoots. This is already mentioned when discussing the results and on reflection we do not need to make such a statement without first showing the results.

L137-138: If we have some initial estimate of the uncertainty for the location of the tipping threshold, then a tipping probability based on this distribution can be assigned for any forcing profile. Make it more clear how the tipping probability is assigned as currently it is unclear the how.

We have moved the text from the appendix to the main text to before introducing Figure 1(c). We now explicitly state that "For any given overshoot profile, there will be a critical threshold location that separates tipping (lower thresholds) from not tipping (higher thresholds). Therefore, the cumulative probability density function at this critical threshold gives the probability of tipping. This probability of tipping, or more precisely the critical tipping threshold location, can either be calculated numerically (using a bisection method) or via a modification to the inverse square law." Here, we have additionally moved the modified theory that accounts for time over an arbitrary threshold to the main text and referred to appendix A1 for the full derivation as to not disrupt the flow too much.

L222: It would be helpful in the beginning of this section to make sure to highlight what is going to be relevant for overshooting scenarios/uncertainty.

We have now added the following sentence to the beginning of the section "In this section we consider how

diffusive timescale uncertainties in a low-dimensional box model for the AMOC affects the uncertainty in tipping behaviour (i.e. if the AMOC collapses or not) for overshoot profiles of the freshwater flux into the North Atlantic.".

L222: I would remind readers that the results/limitations of this section are based on low-dimensional box model, or you can put this into the conclusions.

Please see comment above, where we clearly now state in the opening sentence that the results in the section will be based on a low-dimenisonal box model.

L234: Do results substantially change if the advective timescale is not fixed at 70 years? How was 70 exactly chosen?

The advective timescale was chosen arbitrarily (this has now been added to the manuscript) to be roughly in the middle of the range provided by the literature. The results do not change qualitatively but do change quantitatively a little for a different advective timescale with the range presented.

General comment: Legends in figures for colors/lines would make the figures easier to decipher without needing to read the entire figure caption.

We now include legends for the relevant figure panels.

General comment: Are the equations supposed to be centered or is this just an artifact of the preprint (L94, L320)?

The equations are not currently centred, but this will be sorted upon typesetting.

**Technical comments:**

L3: The sentence beginning with "One common mechanism for tipping to occur is via forcing a nonlinear system..." is awkward. The start of the sentence should be changed to make this sentence flow better.

The sentence has been rephrased to read "Forcing a nonlinear system beyond a critical threshold that signifies the onset of self-amplifying feedbacks provides one possible mechanism for tipping."

L29-30: There is unnecessary commas.

We have now removed the unnecessary comma.

L49: Specify the "those" in: . . . in addition to those, . . .

The "those" referred to the studies but we have now removed the first part of the sentence for conciseness.

L81-82: Say datasets back-to-back, I would suggest rewording.

The second mention of datasets has been removed, such that the revised sentence now reads "One type of uncertainty related to a possible tipping of the AMOC is uncertainty in datasets such as those for sea-surface temperature and salinity."

L120: I suggest losing the parentheses about calling it threshold later and just work that into this sentence. If you are changing notation/how you will refer to things, don't make it a small note.

The opening to this section has been reworked and the original sentence has been deleted. Further, we have added 'tipping' before 'threshold' to avoid any ambiguity where relevant.

L124: This first half of the first sentence reads a little bit awkwardly. Eliminate at least one of the commas if possible.

We have now reformulated the start of the sentence to read "We consider a single forcing profile, which we assume to start at some initial level of forcing, smoothly increases...".

L136: We have seen that for a given overshoot forcing profile. . . refer to where we just saw this profile.

We now refer to Figure 1(a), (b) explicitly by writing, "In Figure 1(a), (b), we have seen that for a given overshoot forcing profile...".

L387-390: This paragraph feels out of place. It should perhaps be earlier in the conclusion.

We see it more as a summarising part of our conclusion. We have reformulated it so that it becomes clearer that our study revealed the high influence of the threshold location and linear restoring force on the uncertainty in the probability of tipping. Constraining these two system characteristics helps to constrain the mitigation window.

---

## Referee Report (RR1)

**Review of manuscript egusphere-2024-2170**

'*Uncertainty quantification for overshoots of tipping thresholds*' by Lux-Gottschalk & Ritchie

The authors have addressed the comments, questions and suggestions included in my first review in their responses and revisions. In particular, restructuring the manuscript by describing the models and the deviation of the tipping probability before presenting the impact of uncertainty in the location of the critical threshold and the linear restoring force on the tipping reponse helps to follow and understand the key results of this study.

I have a few more specific comments, questions and suggestions which, in large parts, are related to the clarity of the language in the revised manuscript. These are listed below.

**Specific comments**

L01 (and L22): I am not sure whether parts of the Earth system (including those potentially displaying tipping dynamics) can be considered to be currently in a stable state. The question that I posed in my first review may have been misleading. Maybe formulate more carefully. See also L432-L344, where this assumption is, in fact, discussed for the AMOC in this manuscript.

L04: Maybe replace "provides one possible mechanism for tipping" by "constitutes a possible mechanism for tipping", as no other mechanism of tipping is described in the abstract.

L09: Maybe replace "look at" by "assess" or "explore".

L09: Maybe replace "affect the probability of tipping" by "propagate to uncertainties in the probability of tipping" (or similar).

L15: I am not sure if "pathway believed to offer little danger of tipping" fits well here. Please consider to reformulate this phrase.

L20: Please consider to also add older references. Tipping dynamics in the Earth system have not just gained attention recently, see, for example, van Nes et al. (2016) and Lenton et al. (2008) (which is also cited at a later point in the manuscript).

L26-L28: I am not sure if this sentence adds any new information compared to the previous sentences, except introducing the AMOC as an example for a tipping system. I would suggest to reformulate along the lines of, for example, "One example for a part of the Earth system suggested to exhibit tipping behaviour is the Atlantic Meridional Overturning Circulation (AMOC)". Please also consider to add a few more specific references on the possible tipping dynamics of the AMOC, e.g. based on Weijer et al. (2019) and references therein.

L29: Maybe replace "trigger" by "lead to".

L32: Maybe replace "impact is" by "impacts are".

L33: What is meant by "measures"? Please consider to be more precise in the wording here.

L34: Overshoots of tipping thresholds have not been introduced so far, but have only been described in the abstract (L05-L07). Please add a short explanation, and a motivation why overshoots should be studied in the introduction.

L36-37: Please consider adding "without a change in the external forcing".

L39: Please consider adding "by (slowly) changing the external forcing".

L43: It could be helpful to explain what you mean by "mitigation window" in the introduction. A possible explanation is given in L86-88 with respect to overshoots, but it is unclear whether / how this applies here.

L43-L45: How does this justification in terms of a "natural restriction" related to L34-35, describing that "it is important to understand which mechanisms can cause a system to tip" to "understand overshoots of tipping thresholds". I think I understand what you mean, but wonder if clarity could be improved here.

L47: Maybe it is possible to clarify this sentence. Do you mean something along the lines of "climate model simulations have suggested that tipping might not occur despite an overshoot"?

L54-55: Maybe consider to rephrase along the lines of "emphasises that high-impact, low-likelihood climate outcomes, to which some tipping events belong, should be part of climate risk assessments", if appropriate.

L57-58: Please consider to briefly describe what "indicators of critical slowing down" refers to for readers that are not familiar with early warning signals.

L60: Maybe replace "retain" by "return to".

L79: Maybe replace "environmental parameter" by "external forcing" for clarity, as used previously in the manuscript and, for example, in the axis labelling in Figure 1.

L79: Maybe replace "have different impacts on the overshoot of the critical bifurcation (threshold) value" by, for example, "have different characteristics with respect to the overshoot of the critical threshold / bifurcation value".

L81-82: I would like to suggest "These uncertainties introduce uncertainties in the time and peak overshoot distance...".

L84: Maybe replace "thus includes a range of forcing parameters where the AMOC exhibits multistability" by "thus there exists a range of external forcings where the AMOC is characterized by multistability".

L86-87: I would suggest to reformulate along the lines of "inverse-square law between the peak overshoots distance, exceedance time and system characteristics".

L98: Previously, "peak external forcing" was described as "peak overshoot distance". I would suggest to stick to one of these terminologies throughout the manuscript for clarity.

L105: "affects the tipping behaviour" or "affects the uncertainty in the tipping behaviour"?

L111-112: Since a general reference to bifurcation theory has already been given in the introduction, this may not be needed here.

L115: To me, this reads like a repetition of the first sentence of this paragraph. Please consider removing or bringing both sentences together.

L116: I am not sure whether the reference point for the distance to the basin boundary is clear. Alternatively, this could maybe be replaced by "the size of the basin of attraction".

L116: Comma may not be needed.

L117: Does "all system characteristics" refer to the linear restoring force and the distance to the basin boundary? Please consider to clarify here.

L117: To me, this formulation is confusing after reading the preceding sentence. Maybe replace "remain unchanged" by "are the same".

L117: What is meant by "system's"? Does this refer to (two) systems that differ in terms of the location of their tipping threshold (that is "systems")? Or the "systems' states"?

L117-118: If I understand correctly, the basins of attractions, however, have different sizes for the two example systems considered here, when the external forcing approaches zero again after the overshoot (and the systems are not at the same distance to their respective thresholds). Does the argument described in L193-L196 for the case of the uncertain linear restoring force also play out for uncertainties in the location of the tipping threshold?

L119: Maybe consider to remove "develop this idea further".

L124: Maybe replace "for less time" by "of shorter duration" (or similar).

L125: If possible, please formulate "contrasting consequences" more specific.

L127: "curves show" instead of "curvesshow"

L135: Are "the probability of tipping" and the "critical tipping threshold location" equivalent? Maybe also consider to find another phrase for "critical tipping threshold location". I am not sure whether this phrase is intuitive, given that it could be understood as the critical threshold of an example system, while (if I understood correctly) it here, for a given overshoot profile, refers to *the* critical threshold of *the* system that separates tipping from not tipping (L133).

L158: Please consider to reformulate along the lines of "For lower thresholds, however, an overshoot of the threshold..."

L159: Maybe replace "larger peaks" by "larger peaks in the external forcing" or similar.

L164: If I understood correctly, in L126-L127 the same trajectory is described to "tip due to the large and long overshoot". This formulation may seem a bit contradictory to "tipping is nearly avoided". Please check the formulation.

L172: Please consider to expand this sentence to improve clarity, for example, as "For example, forcing profiles that were very unlikely to result in tipping (10% probability of tipping) for

the initial distribution are now exceptionally unlikely (< 1%) to give rise to a critical transition given the knowledge-based distribution."

L180: After introducing Section 2, this explanation of the strength of the linear restoring force may not be needed here. I would suggest to move it to Section 2, where this quantity is formally introduced, or to integrate into the sentence for better readability.

L189: Please consider to expand this sentence for clarity, for example, as "... does not cross the unstable branch (representing the boundary of the basin of attraction) when reducing the external forcing...", if appropriate.

L193: I think I understand what is meant here, but would suggest to reformulate. I am not sure whether a system can "take longer to realise it is over the edge"?

L193-194: I would suggest to reformulate along the lines of "In addition, for a weaker restoring force proportionality factor, the boundary of the basin of attraction is further away from the initial stable state (blue dot in Fig. 3b)".

L202: Maybe replace "then plots" by "shows".

L203: Maybe replace "not unlike before this is now fixed" by, for example, "which is now fixed, in contrast to Sect. 3.1".

L203-205: I think this could be removed here, and should rather be part of the figure caption.

L212: Maybe replace "whereas" by "in contrast".

Figure 4: I was wondering whether Figure 4b might be easier to understand if it would be split into two panels, showing the tipping probability for (1) the initial and knowledge-based distribution of the restoring force probability factor, and (2) the initial and alternative distribution of the restoring force probability factor?

L244: Please integrate "a proxy for the strength of the AMOC" into the sentence to improve readability.

L264-267: If I understand correctly, these sentences are also relevant for the description of Figure 5c (L281-LL283). Would it be possible to bring these parts together?

L288: How do "arbitrarily" and "the advective timescale is relatively well constrained" / "within the reasonable physical range" fit together? Please clarify or reformulate.

L290: If I understand correctly, the red dashed line indicates the value chosen for the advective timescale (see caption of Figure 5). Please move the reference to the red dashed lines accordingly (e.g. to L288), or clarify.

L316: To me, "We now again follow a similar approach as for Figures 1 and 3" reads like a repetition of the first sentence of this paragraph. Please consider removing or bringing both sentences together.

L345-346: I would suggest to phrase this sentence the other way around for clarity, i.e. "The large uncertainty in the system parameter, here in terms of the diffusive timescale, again causes large uncertainties in the tipping behaviour".

L359: Please consider specifying as "posterior distribution of the diffusive timescale".

L359: Please define "mitigation window", see previous related comment.

L365-366: Please consider simplifying this sentence.

L384-388: If I understand correctly, this section describes the effects of an increasing standard deviation of the normally distributed diffusive timescale on the tipping uncertainty (resulting in a larger tipping uncertainty), while L 393-401 describes the effects of a decreasing standard deviation (potentially leading to a reduction in the uncertainty of tipping behaviour). Please consider to bring both sections together.

L398-L399: Please add a reference to Figure 8a for clarity.

L405: Maybe remove "individually" or reformulate. "isolated" and "individually" may not be needed together.

L458: Maybe reformulate along the lines of "the strong influence of the uncertainty in the tipping threshold location".

L459: Maybe replace "seen" by "shown".

L461: Maybe replace "if we want to avoid the tipping of elements of the climate system" by "to avoid critical transitions in parts of the climate system" or "to avoid tipping in parts of the climate system".